# Cleavage activates Dispatched for Sonic Hedgehog ligand release

**Daniel P Stewart[1†], Suresh Marada[1†], William J Bodeen[1,2], Ashley Truong[1], Sadie Miki Sakurada[1,3], Tanushree Pandit[1], Shondra M Pruett-Miller[1,3], Stacey K Ogden[1]\***

[1]Department of Cell and Molecular Biology, St. Jude Children's Research Hospital, Memphis, United States; [2]Integrated Program in Biomedical Sciences, University of Tennessee Health Sciences Center, Memphis, United States; [3]Center for Advanced Genome Engineering, St. Jude Children's Research Hospital, Memphis, United States

**Abstract** Hedgehog ligands activate an evolutionarily conserved signaling pathway that provides instructional cues during tissue morphogenesis, and when corrupted, contributes to developmental disorders and cancer. The transmembrane protein Dispatched is an essential component of the machinery that deploys Hedgehog family ligands from producing cells, and is absolutely required for signaling to long-range targets. Despite this crucial role, regulatory mechanisms controlling Dispatched activity remain largely undefined. Herein, we reveal vertebrate Dispatched is activated by proprotein convertase-mediated cleavage at a conserved processing site in its first extracellular loop. Dispatched processing occurs at the cell surface to instruct its membrane re-localization in polarized epithelial cells. Cleavage site mutation alters Dispatched membrane trafficking and reduces ligand release, leading to compromised pathway activity in vivo. As such, convertase-mediated cleavage is required for Dispatched maturation and functional competency in Hedgehog ligand-producing cells.
DOI: https://doi.org/10.7554/eLife.31678.001

**\*For correspondence:**
stacey.ogden@stjude.org

[†]These authors contributed equally to this work

**Competing interests:** The authors declare that no competing interests exist.

## Introduction

Hedgehog (Hh) ligands are produced as precursor proteins that undergo autocatalytic processing whereby a carboxyl-terminal intein-like domain cleaves itself, in a cholesterol-dependent manner, from an amino-terminal signaling domain (*Guy, 2000*; *Lee et al., 1994*). The resulting ~20 kDa signaling protein is covalently modified by cholesterol on its new carboxyl-terminal cysteine and by a long chain fatty acid on its amino-terminus (*Porter et al., 1996a*; *Chamoun et al., 2001*; *Pepinsky et al., 1998*; *Long et al., 2015*). These lipid modifications contribute to physiological Sonic Hh (Shh) activity by governing ligand distribution across developing tissues, and influencing ligand potency toward target cells (*Long et al., 2015*; *Taylor et al., 2001*; *Li et al., 2006*; *Porter et al., 1996b*; *Burke et al., 1999*). Distribution is controlled by Shh lipid modifications conferring high membrane affinity to the mature ligand, thereby anchoring it to the producing cell surface. In order to reach long-range target cells, Shh must be deployed from producing cell membranes through a process that is dependent upon Dispatched 1 (Disp), a predicted twelve-pass transmembrane protein that shares homology with the bacterial Resistance, Nodulation and Division (RND) Transporter superfamily (*Burke et al., 1999*; *Caspary et al., 2002*; *Ma et al., 2002*; *Kawakami et al., 2002*; *Amanai and Jiang, 2001*). *Disp1* knockout mice phenocopy animals lacking the essential Shh signal transducing component Smoothened (Smo), underscoring the importance of Disp for pathway activity during early development (*Caspary et al., 2002*; *Ma et al., 2002*; *Kawakami et al., 2002*).

**eLife digest** As an embryo develops, its cells divide many times until they specialize to become distinct cell types that make up the tissues and organs. To do so, the cells need to communicate, and some send signals by making and releasing proteins that travel to more distant cells. One such signaling pathway is called Hedgehog signaling. This pathway is necessary so that the tissue and organs develop properly. If faulty, it can stop the embryo from developing properly and even lead to diseases such as cancer.

Hedgehog signaling is initiated by the Hedgehog protein, which needs to be released from the cells that produce the message to transport the signal to the target cells. A protein called Dispatched helps Hedgehog to get free and travel to its destination. Without Dispatched, Hedgehog cannot be released and the embryos will not develop. Now, Stewart, Marada et al. wanted to find out if and how Dispatched itself is controlled by studying embryo cells of mice.

The results showed that a protein called Furin activates Dispatched by splitting it at a specific point. When the break-point on Dispatch was genetically modified, Furin could no longer cleave the protein. As a consequence, Dispatched did not reach the correct location within cells to help Hedgehog move away from signal-releasing cells.

This suggests that Furin is an essential protein of the Hedgehog pathway. A next step will be to see if this is also the case in humans. Some cancer cells can produce large amounts of Hedgehog protein, which makes tumors grow faster. A better understanding of Hedgehog signaling may help to find new cancer therapies that can block this pathway in cancer cells.

DOI: https://doi.org/10.7554/eLife.31678.002

In vertebrates, Disp functions with the secreted glycoprotein Scube2 to facilitate Shh membrane extraction (*Ma et al., 2002*; *Creanga et al., 2012*; *Tukachinsky et al., 2012*). The precise mechanism by which Disp and Scube2 mobilize Shh from the producing cell membrane is not yet clear. However, Disp contains a sterol sensing domain (SSD) that is thought to interact with the Shh cholesterol modification to position the ligand for transfer to Scube2 (*Creanga et al., 2012*; *Tukachinsky et al., 2012*). Despite this advance in understanding the Disp-Scube2 functional relationship, little is known about how Disp activity is regulated. Biochemical and cell biological analyses have shown Disp must organize into trimers and localize to the basolateral cell surface to release Shh (*Etheridge et al., 2010*). Genetic studies in *Drosophila* suggest a crucial role for Disp-mediated endosomal recycling during Hh deployment, demonstrating that apically localized Hh must be internalized in a Disp-dependent manner, and then retargeted to the cell surface to exit ligand-producing cells (*D'Angelo et al., 2015*; *Callejo et al., 2011*). Loss of Disp function triggers apical accumulation of Hh and disruption of long-range signaling (*D'Angelo et al., 2015*; *Callejo et al., 2011*), suggesting the ability of Disp to appropriately traffic with Hh is imperative for ligand release. The regulatory processes influencing Disp membrane targeting and recycling have not yet been established.

Herein, we demonstrate that Disp membrane targeting and recycling is dependent upon convertase-mediated cleavage. Cleavage occurs at an evolutionarily conserved site in the predicted first extracellular loop of Disp (EC1) by the proprotein convertase Furin. Mutation of the EC1 cleavage site prevents Disp processing and disrupts Shh deployment, consistent with convertase cleavage being an essential step in Disp functional maturation. Results suggest that Disp is clipped at the cell surface and that the resulting amino-terminal fragment and processed carboxyl domain are differentially trafficked post-processing. Disruption of processing by cleavage site mutation results in altered membrane distribution of Disp, leading to compromised pathway activity in vivo. Combined, these results establish cleavage as an essential step for Disp functionality, and provide novel mechanistic insight into control of Disp function in ligand-producing cells.

## Results

To begin biochemical and cell biological analysis of Disp regulation, we generated a carboxyl-terminally HA epitope-tagged murine Disp (DispHA) expression vector. All commercial and custom anti-Disp antibodies tested failed to detect the murine Disp protein, necessitating use of the epitope-tagged expression vector. Western blot of cell lysates from NIH3T3 cells transfected with plasmid

encoding DispHA revealed two distinct protein bands detected by anti-HA antibody, one running near the predicted molecular weight of 175 kDa, hereafter referred to as Disp175, and a second with an apparent molecular weight of ~145 kDa, Disp145 (*Figure 1A*). Because membrane and secreted proteins are commonly modified by addition of N-linked glycans, we tested whether the size difference of the two species resulted from differential N-glycan modification. Lysates from cells expressing DispHA were treated with Endo H or PNGase F enzymes, and their migration on SDS-PAGE gels was assessed. Treatment with Endo H, which removes simple N-glycans added in the endoplasmic reticulum (ER), resolved a Disp protein species from Disp175, indicating a fraction of the upper band was ER-localized (*Figure 1B* lane 2, arrowhead). The lower band was resistant to Endo H. However, PNGase F, which strips both simple and complex post-ER glycans, significantly altered migration of Disp145, indicating post-ER localization of the smaller protein species (lane 3,

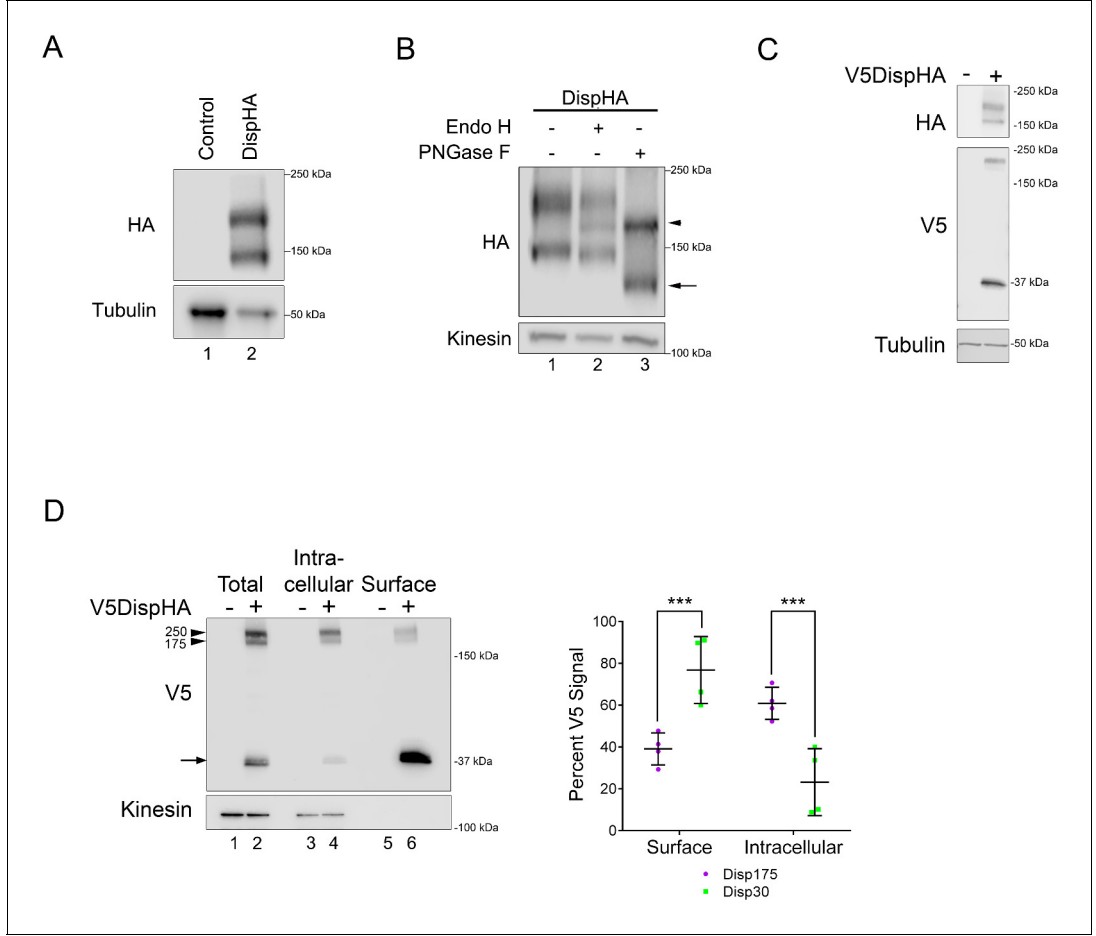

**Figure 1.** Disp is cleaved. (**A**) Lysates prepared from vector control and DispHA-expressing NIH3T3 cells were analyzed by SDS-PAGE and western blot against the HA tag. (**B**) Lysates from DispHA-expressing NIH3T3 cells were treated with Endo H or PNGase F enzymes. The arrowhead marks deglycosylated Disp175 and the arrow marks deglycosylated Disp145. (**C**) V5DispHA was expressed in NIH3T3 cells and cell lysates analyzed by western blot. (**D**) Lysates were prepared from DispHA-expressing NIH3T3 cells treated with biotin-containing culture medium for 30 min at 4°C prior to lysis. Lysates were incubated with streptavidin beads and bound (surface) and unbound (intracellular) DispHA proteins were analyzed by western blot. Combined densitometry analysis of four independent labeling experiments is shown. Densitometry is presented as percentage of the sum total of Disp175 or Disp30 signals across the two fractions. Significance was determined using a paired Student's t-test. For all statistical analyses *p≤0.05 and ***p≤0.005. Error bars indicate standard deviation (s.d.). For all western blots Kinesin or Tubulin serve as loading controls.

DOI: https://doi.org/10.7554/eLife.31678.003

The following source data is available for figure 1:

**Source data 1.** Data for *Figure 1D*.

DOI: https://doi.org/10.7554/eLife.31678.004

arrow). PNGase F treatment collapsed Disp175 to a size similar to its Endo H-sensitive fraction, consistent with the larger protein species containing both ER and post-ER fractions (lane 3, arrowhead).

The observation that the Disp145 fraction was highly enriched for EndoH-resistant glycosylation, suggested Disp175 might be cleaved to generate a truncated protein after ER exit. To test this, a V5 epitope tag was inserted in the amino-terminal region of the predicted EC1, as determined using TMPred and HMMTOP 2.0 secondary structure prediction tools (*Figures 1C* and *2B*). The V5 insertion site was chosen based upon the apparent molecular weight difference of the two DispHA protein species. Double tagged V5DispHA was expressed in NIH3T3 cells, and cell lysates were assessed by western blot. Disp175 was detected by both HA and V5 antisera (*Figure 1C*). Conversely, Disp145 was detected only by HA, suggesting loss of the V5 epitope from the Disp145

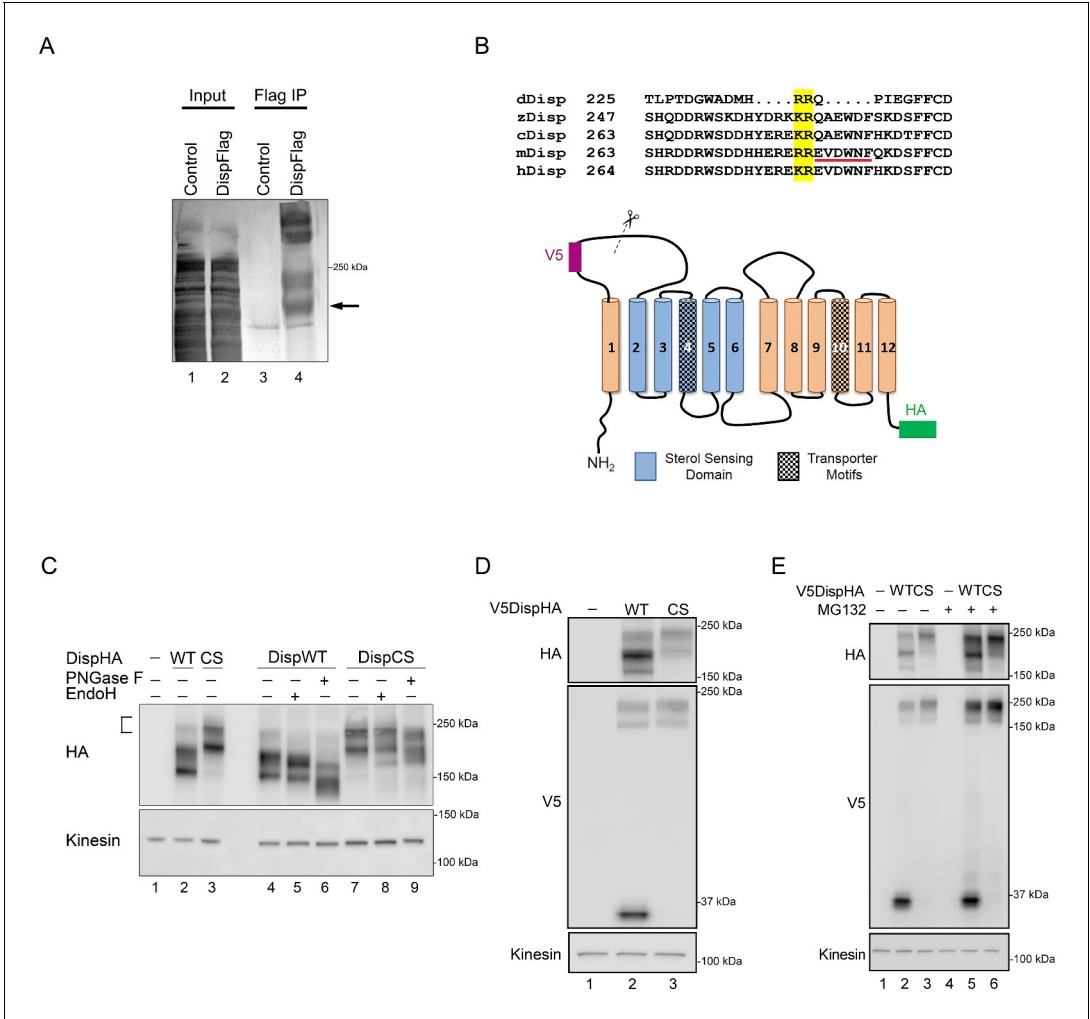

**Figure 2.** Disp is cleaved at a Furin consensus sequence in EC1. (**A**) Disp-Flag was expressed in HEK293T cells, immunopurified from cell lysates on Flag beads and the 145 kDa species (arrow) excised and analyzed by Edman degradation. (**B**) Alignment of EC1 sequence from Disp proteins of *Drosophila*, zebrafish, chick, mouse and human. Edman sequencing of murine Disp145 revealed EVDWNF (red line) to be the amino terminal sequence, suggesting that cleavage occurs adjacent to dibasic residues R279 and E280 (yellow box). A diagram of murine Disp shows V5 and HA epitope tag insertion sites, functional domains and the approximate location of the identified cleavage motif in predicted EC1 (scissors). (**C**) Cell lysates from NIH3T3 cells expressing wild type or cleavage site mutant (CS, R279A/E280A) were treated with deglycosylating enzymes. The bracket indicates the ~250 kDa fraction. (**D**) Lysates from NIH3T3 cells expressing wild type or cleavage site mutant V5DispHA proteins were examined for presence of the 30 kDa V5 fragment by western blot. (**E**) NIH3T3 cells expressing wild type or CS mutant Disp proteins were treated for ~6 hr with DMSO vehicle or MG132 proteasome inhibitor. Kinesin is the loading control.
DOI: https://doi.org/10.7554/eLife.31678.005

species. Accordingly, a ~ 30 kDa fragment was detected by V5 antisera, confirming that Disp protein is clipped to produce Disp145.

To determine whether Disp175 was processed before or after reaching the plasma membrane, cell surface biotinylation experiments were performed (*Figure 1D*). V5DispHA-expressing NIH3T3 cells were incubated in biotin-containing culture medium for 30 min at 4°C prior to lysis, and biotinylated proteins were captured from cell lysates on streptavidin-coated beads. Bound (surface) and unbound (intracellular) fractions were examined by western blot against the V5 tag to detect the unprocessed protein and the ~30 kDa V5 cleavage fragment. Combined densitometry analysis of four independent experiments revealed that unprocessed Disp175 enriched in the non-biotinylated intracellular fraction. Approximately 60% of total Disp175 signal was detected in the unlabeled intracellular fraction with ~40% present on the cell surface (*Figure 1D*. lane 4 vs. 6, arrowhead and densitometry summary, purple). Although Disp175 represented the lesser pool of surface-labeled V5DispHA, its presence on the surface argued against Disp145 conversion occurring prior to it reaching the plasma membrane. Consistent with this hypothesis, the processed 30 kDa V5Disp fragment was significantly enriched on the cell surface, accounting for ~76% of the total V5Disp30 signal (Figure 4D, arrow and densitometry summary, green). Combined with the above deglycosylation analysis, these results suggest Disp145 is likely generated from Disp175 at the cell surface. Its generation in the ER or Golgi is unlikely given the low percentage (~23%) of Disp30 in the intracellular, non-biotinylated fraction.

We next sought to identify the exact cleavage site in EC1. To do so Disp-Flag was expressed in HEK293T cells, and Disp175 and 145 proteins were purified on Flag beads. Disp145 was excised and subjected to Edman degradation to identify its amino-terminal residues (*Figure 2A*, arrow). EVDWNF, which maps to amino acids 280–285 in EC1 of the murine protein, was identified as the amino-terminal sequence (*Figure 2B*, red line). This is directly adjacent to a dibasic amino acid proprotein convertase (PC) cleavage motif that is conserved in Disp proteins from *Drosophila*, zebrafish, chick, mouse and human (*Figure 2B*, yellow box) (*Seidah et al., 2013*). Consistent with this being a functional cleavage site, disruption of the murine Disp convertase motif by R279A, E280A mutation (DispCS) appreciably reduced Disp175 to Disp145 conversion, evidenced by nearly undetectable Disp145 or Disp 30 signal in DispCS cell lysates (*Figure 2C*, lane 3 and 2D, lane 3). The 175 kDa fraction of DispCS was predominantly Endo H resistant, indicating that failure to cleave was not likely due to the mutant protein being retained in the ER (*Figure 2C*, lane 8). Notably, CS mutation resulted in pronounced accumulation of a large molecular weight band (~250 kDa) that was also evident for the wild-type protein, albeit at reduced intensity (*Figure 2C* lane 3 compared to 2, bracket). The 250 kDa fraction of DispCS was largely resistant to deglycosylating enzymes, suggesting the molecular weight shift was not due to alteration of N-glycosylation status (*Figure 2C*, lanes 7–9). Moreover, Disp250 accumulated for both wild type and CS DispHA proteins at equal intensities following proteasome inhibition by MG132 treatment (*Figure 2E*). As such, Disp250 may represent a Disp175 species marked for proteosomal degradation by post-translational modifications such as ubiquitination, neddylation or sumoylation.

To confirm a proprotein convertase was responsible for Disp processing, DispHA-expressing NIH3T3 cells were treated with cell-permeable Furin Inhibitor I, which blocks activity of convertase family members Furin, PCSK1, PCSK2, PACE4, PCSK5 and PCSK7. Treatment with increasing concentrations of drug dose-dependently reduced Disp145 levels (*Figure 3A*, and green in bottom panel). Despite this, steady state levels of the Disp175 precursor species did not increase (magenta). Instead, as was observed following mutation of the cleavage site (*Figure 2C*), chemical inhibition of Furin family proteases triggered accumulation of the ~250 kDa fraction (*Figure 3A*, bracket, and bottom panel, black). Conversion of Disp175 to this larger species likely accounts for the lack of Disp175 accumulation following cleavage inhibition.

Furin inhibitor I sensitivity, combined with results suggesting cleavage occurs after Disp reaches the cell surface (*Figure 1*), indicated a dibasic amino acid-specific convertase such as Furin, PCSK5, PACE4 or PCSK7 (*Seidah et al., 2013*; *Seidah and Prat, 2012*). To identify the specific PC facilitating Disp cleavage, CRISPR/Cas9 technology was used to generate knockout MEF lines for each of these genes. V5DispHA was expressed in two independent clonal lines knocked out for each of these genes, and examined for cleavage by western blot for the 30 kDa V5 fragment (*Figure 3B* and *Figure 3—figure supplement 1*). Knockout of Furin, which targets substrates in the *trans*-Golgi, at the cell surface and in recycling endosomes (*Seidah et al., 2013*), blocked formation of V5Disp30.

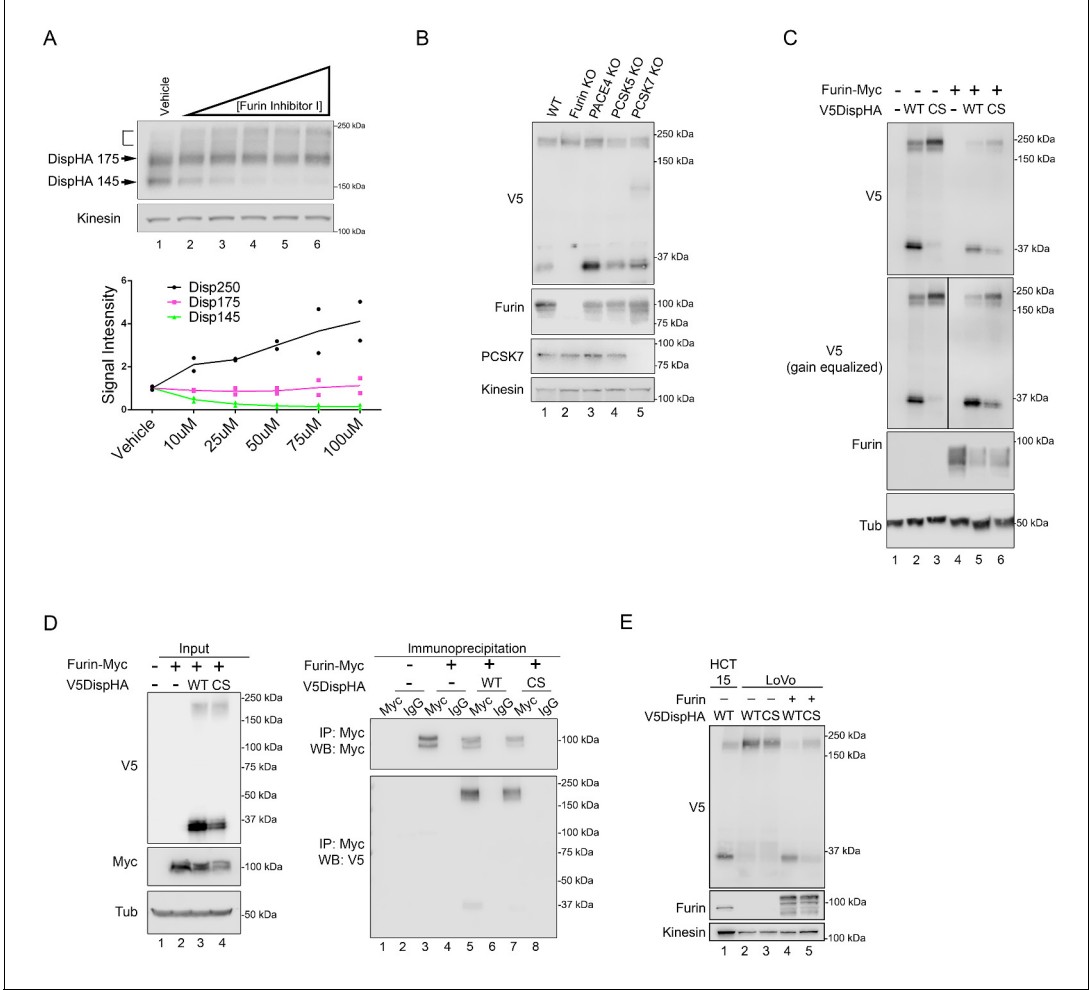

**Figure 3.** Disp is cleaved by Furin. (**A**) NIH3T3 cells expressing DispHA were treated with increasing concentrations of Furin Inhibitor I (10, 25, 50, 75 and 100 μM). The bracket indicates the 250 kDa fraction. Graph shows Disp densitometry analysis normalized to Kinesin for two independent experiments. Normalized signal intensity for each DispHA species in treated conditions is shown relative to vehicle control intensity, which was set to 1. (**B**) CRISPR/Cas9 generated knockout lines for Furin, PACE4, PCSK5 and PCSK7 were transfected with V5DispHA-expression vector, and formation of the ~30 kDa V5 cleavage fragment was monitored by western blot of cell lysates from Clonal line 1. Furin and PCSK7 protein levels were examined by western blot. PACE4 and PCSK5 mutations were confirmed by deep sequencing as in *Figure 3—figure supplement 1*. (**C**) Lysates from cells co-expressing wild type or CS mutant V5DispHA and Furin-Myc proteins were examined for Disp cleavage by western blot for the V5 fragment. Co-expression of Furin-Myc reduced total Disp signal (top panel). Gain equalization of the V5 signal in Furin-Myc expressing cell lysates is shown for comparison. (**D**) Furin-Myc and V5DispHA were co-expressed in HEK293T cells and Furin-Myc was immunoprecipitated from lysates using anti-Myc. Input (left) and immunoprecipitates (right) are shown. (**E**) Wild type and CS mutant V5DispHA proteins were expressed in LoVo (lacking Furin) or HCT-15 (control) colon carcinoma cells and lysates were analyzed by western blot. Re-expression of Furin in LoVo cells rescued cleavage (lane 4 compared to 2). Kinesin and Tubulin are the loading controls for western blots.

DOI: https://doi.org/10.7554/eLife.31678.006

The following source data and figure supplement are available for figure 3:

**Source data 1.** Data for *Figure 3A*.

DOI: https://doi.org/10.7554/eLife.31678.008

**Figure supplement 1.**

DOI: https://doi.org/10.7554/eLife.31678.007

Knockout of PCSK5, PACE4 and PCSK7 did not, identifying Furin as the candidate convertase responsible for Disp cleavage. Accordingly, over-expression of epitope-tagged Furin-Myc with V5DispHA enhanced cleavage of the wild-type protein, and induced low-level cleavage of the CS mutant, suggesting that by increasing Furin protein levels compensatory and/or off-site cleavage can occur (*Figure 3C–D*). To test for Furin-Disp association, V5DispHA was co-expressed with Furin-Myc

in HEK293T cells and anti-Myc immunoprecipitation experiments were performed (*Figure 3D*). Both wild-type and CS mutant V5DispHA proteins were detected in anti-Myc immunoprecipitates, consistent with an interaction occurring between Furin and Disp (right panel, lanes 5 and 7). V5DispHA was not collected by anti-Myc in the absence of Furin-Myc expression, confirming specificity of the immunoprecipitation (right panel, lanes 1–4).

To further test for a specific requirement for Furin in facilitating Disp cleavage, V5DispHA was expressed in Furin-deficient colorectal adenocarcinoma-derived LoVo cells, and generation of the V5Disp30 cleavage fragment was assessed (*Takahashi et al., 1993*). When expressed in control HCT-15 colorectal cells, both 175 kDa and 30 kDa protein species were evident (*Figure 3E*, lane 1). Conversely, murine V5DispHA protein expressed in LoVo cells failed to produce V5Disp30, and instead migrated in a manner similar to the CS mutant (lanes 2–3). Cleavage disruption was specific to Furin loss because its re-expression in LoVo cells rescued V5Disp30 production (lane 4). Combined with the above, these results support that Disp is cleaved by Furin.

Proprotein convertases such as Furin typically act on inactive proproteins to remove inhibitory or regulatory domains as a requisite step in functional maturation of the substrate. However, a small number of substrates are inactivated by convertase cleavage (*Seidah et al., 2013*). To test how Furin cleavage affected Disp, Shh transcriptional reporter assays were performed by co-culturing Shh-responsive LightII reporter cells with *Disp-/-* mouse embryonic fibroblasts (MEFs) engineered to stably express Shh (*Ma et al., 2002*; *Taipale et al., 2000*). Shh-expressing *Disp-/-* MEFs were transiently transfected with vectors encoding GFP control, wild-type DispHA, DispHA-CS or a published nonfunctional Disp mutant, DispHA-TM. This mutant harbors mutations of conserved residues in the predicted transporter motifs in TM domains 4 and 10 (*Figure 2B* [*Ma et al., 2002*]). Ligand-producing cells were co-cultured with LightII reporter cells for ~48 hr, and reporter induction was measured (*Figure 4A*). Co-culture of LightII cells with *Disp-/-* MEFs expressing GFP +Shh failed to induce a significant change in reporter gene activity over that of the GFP control. Conversely, co-culture of reporter cells with Shh-expressing MEFs transiently expressing wild-type DispHA induced a statistically significant reporter response, consistent with Shh deployment being rescued by re-expression of wild-type DispHA protein. Expression of DispCSHA in Shh-expressing *Disp-/-* MEFs affected LightII cell reporter activity similarly to the GFP control. This reduced signaling level was also similar to what was observed in LightII cells co-cultured with Shh-stable *Disp-/-* MEFs expressing nonfunctional DispTM-HA (*Ma et al., 2002*). As such, attenuation of Disp cleavage likely compromises Shh deployment to target cells.

To directly test the ability of DispCS to deploy Shh, ligand release into culture media of Shh-expressing *Disp-/-* MEFs was examined. In vertebrates, Disp functions with the secreted glycoprotein Scube2 to promote ligand release (*Creanga et al., 2012*; *Tukachinsky et al., 2012*). Therefore, wild type or DispCSHA proteins were co-expressed with increasing Scube2-Flag in Shh-stable *Disp-/-* cells, and ligand accumulation in culture media was monitored by western blot and densitometry analysis of protein-normalized media samples (*Figure 4B*). In the absence of Scube2, Shh was not detected in culture media of GFP and Shh-expressing *Disp-/-* cells (*Figure 4B*, lane 2, media and light gray in densitometry analysis). Co-expression of Scube2-Flag was unable to bolster Shh release from GFP-transfected cells, despite efficient Scube2-Flag secretion from the Shh-stable *Disp-/-* cells (lanes 3–5, media and light gray). Low-level re-expression of wild-type DispHA in MEFs modestly increased Shh release into culture media over that of the GFP control (lane 7 compared to 2, media and dark gray in densitometry analysis). Consistent with Scube2 partnering with Disp to facilitate ligand extraction from the membrane, co-expression of increasing levels of Scube2-Flag with DispHA prompted a dose-dependent increase in Shh protein detectable in conditioned culture media (lanes 8–10 and dark gray). Conversely, cleavage-deficient DispCS failed to effectively promote Shh release into conditioned media when expressed alone or in combination with Scube2-Flag (*Figure 4B*, lanes 12–15 and black). Shh release was similarly affected by genetic elimination of Furin (*Figure 4C*). Whereas control MEFs released Shh into culture media, CRISPR/Cas9 generated *Furin-/-* MEF clones, which failed to effectively cleave V5DispHA, were compromised in their ability to release ligand (lanes 1–2 vs 3–4). Combined, these results support that Disp cleavage is necessary for Shh deployment.

To assess whether cleavage site disruption would compromise Disp activity in vivo, we turned to the *Drosophila* system, which is a robust and genetically tractable model for Hh signal transduction (*Jiang and Hui, 2008*; *Lee et al., 2016*). We first confirmed processing of endogenous *Drosophila*

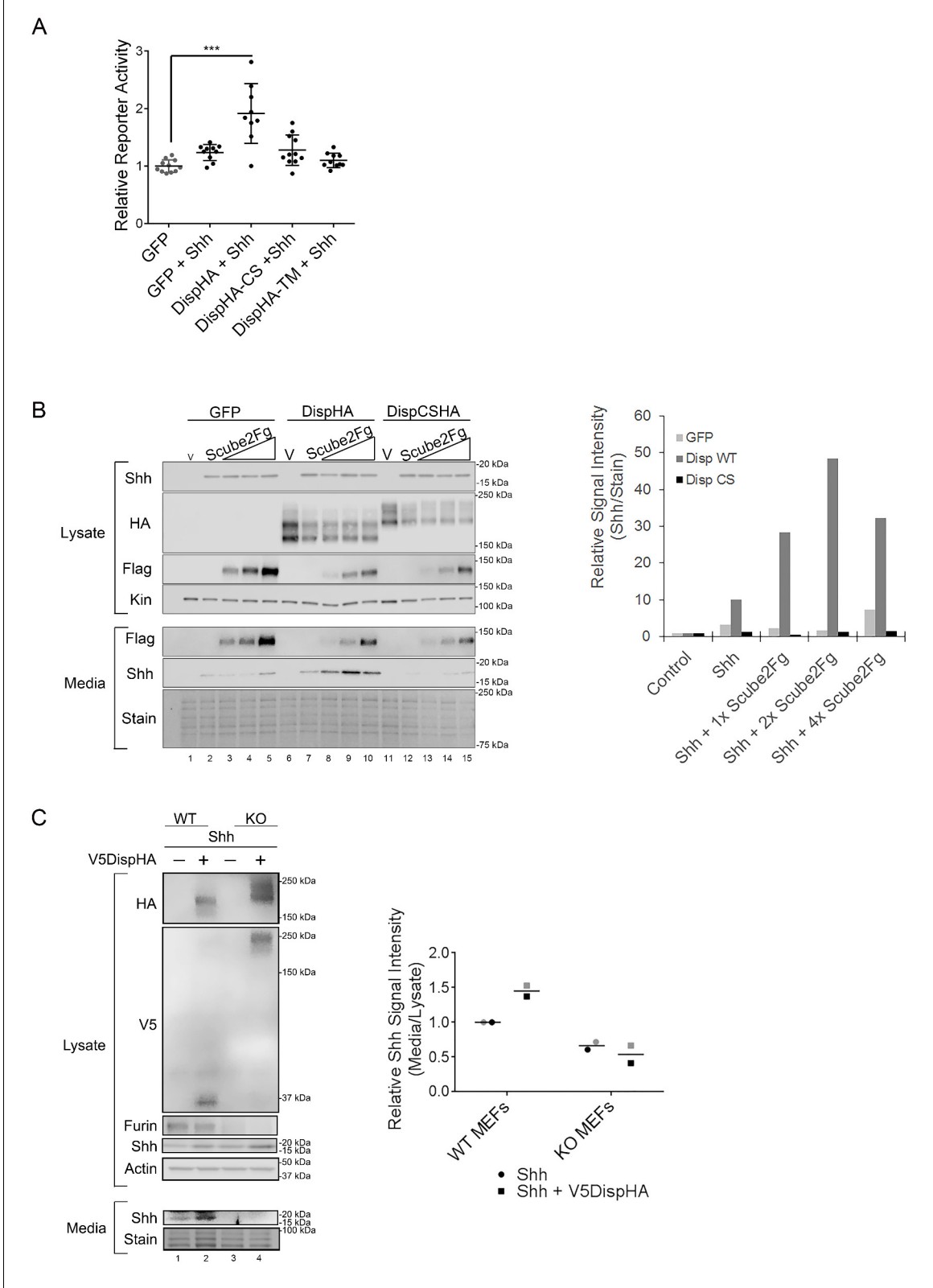

**Figure 4.** Disp cleavage is required for Shh release. (**A**) *Disp-/-* cells stably transfected with Shh or empty vector control were transiently transfected with GFP or the indicated DispHA expression vectors, and then co-cultured with LightII reporter cells. Reporter activity normalized to *tk-renilla* and relative to GFP control (set to 1) is shown. Assays were performed four times in duplicate or triplicate and all data points pooled. Error bars represent s.d. Significance was determined by one-way ANOVA. p***≤0.005. (**B**) *Disp-/-* mouse embryonic fibroblasts were stably transfected with empty vector

*Figure 4 continued*

control (V) or vector encoding Shh. Wild type or CS mutant Disp proteins were transiently expressed in *Disp-/-* cells alone or with Scube2, and lysates and media were examined by SDS-PAGE and western blot. Equal protein amounts (25 µg) from TCA precipitates of conditioned media were analyzed. Kinesin serves as loading control for lysate. Coomassie stain of membrane is shown as loading control for conditioned media (bottom). The graph (right) represents densitometry analysis of Shh media signal intensity normalized to media coomassie stain. The experiment was repeated three times. A representative experiment is shown. (C) *Furin-/-* cells were transiently transfected with Shh alone or with V5DispHA expression vector, and tested for Shh release as in (B). The experiment was repeated twice for *Furin-/-* Clone 2 and once for Clone 1, and a representative blot for *Furin-/-* Clone 2 is shown. The graph (right) represents densitometry analysis of Shh media signal normalized to media coomassie stain, and plotted relative to lysate Shh signal normalized to Actin. Analysis of the representative Clone 2 western is black. Densitometry analysis of the corresponding Clone 1 release assay is shown in gray.

DOI: https://doi.org/10.7554/eLife.31678.009

The following source data is available for figure 4:

**Source data 1.** Data for *Figure 4A*.
DOI: https://doi.org/10.7554/eLife.31678.010
**Source data 2.** Data for *Figure 4B*.
DOI: https://doi.org/10.7554/eLife.31678.011
**Source data 3.** Data for *Figure 4C*.
DOI: https://doi.org/10.7554/eLife.31678.012

Disp (dDisp) in cultured fly cells using a polyclonal antibody raised against predicted EC4 of dDisp (*Figure 5A–A′*). Wing imaginal disc-derived Clone 8 (Cl8) cells were treated with control or *5'dispUTR* dsRNA to assess endogenous dDisp, or transfected with a dDispHA expression vector to assess over-expressed protein. The dDisp antibody detected two distinct bands, one migrating at the predicted molecular weight of ~150 kDa (dDisp150) and a second species with an approximate molecular weight of ~110 kDa (dDisp110). The intensity of both bands decreased following *disp* dsRNA treatment and increased with over-expression of epitope-tagged dDispHA (*Figure 5A*). However, the ratio of the fractions shifted from being approximately equal for the endogenous protein to the upper band being predominant when over-expressed (*Figure 5A*, lane 1 vs. 3). Similar to what was observed for mouse Disp, a ~ 30 kDa V5 fragment was released from double-tagged V5dDispHA expressed in Cl8 cells (*Figure 5B*). To test for processing of endogenous dDisp protein in vivo, dDisp was immunoprecipitated from wing imaginal disc lysate prepared from third instar larvae using the dDisp antisera (*Figure 5A′*). Both bands were evident at equal levels in anti-Disp immunoprecipitates, but not in IgG control immunoprecipitates, confirming that endogenous dDisp protein is processed in cultured fly cells and in vivo in wing imaginal discs.

In addition to the dibasic convertase cleavage motif at 237/238 of the fly protein that aligns with the mouse cleavage site (*Figure 2B*), we identified additional consensus motifs at amino acids 209 and 218. Mutation of each of the three sites on their own did not block cleavage (not shown). We therefore engineered an in frame deletion to remove sequence encompassing all three putative cleavage sites (Δ206–238), and tested for processing of the ΔCS mutant in Cl8 cells. Deletion of the three putative cleavage sites ablated generation of dDisp110 (*Figure 5C* lane 4 compared to 1). Endo H and PNGase F sensitivity analysis revealed that like the mouse protein, dDisp110 of the wild type protein harbored complex N-linked glycans, indicative of post-ER localization (*Figure 5C*, lane 3). The dDispΔCS mutant showed both ER and post-ER fractions, indicating loss of cleavage did not result from ER retention (*Figure 5C* lanes 4–6).

Hh patterns the *Drosophila* wing by controlling gene expression in the wing imaginal disc (*De Celis, 2003*). Alteration of Hh signaling during wing development triggers phenotypes in the adult wing, providing a robust system for monitoring changes in pathway activity in vivo. Wild type or ΔCS *UAS-dispHA* transgenes were expressed in the dorsal compartment of wing imaginal discs using the *apterous-GAL4* driver, and adult wings were screened for phenotypes. Consistent with the established positive role of dDisp in Hh release (*Burke et al., 1999*), over-expression of wild-type dDispHA triggered obvious blistering of the adult wing (*Figure 5E* compared to D). This phenotype is similar to what is observed in response to dorsal wing disc over-expression of activating mutants of the Hh signal transducing component Smo (*Marada et al., 2013*). Blistering is indicative of over-growth of the dorsal face of the wing blade, likely resulting from over-proliferation of dorsal compartment cells in response to enhanced Hh release by over-expressed dDispHA. By comparison,

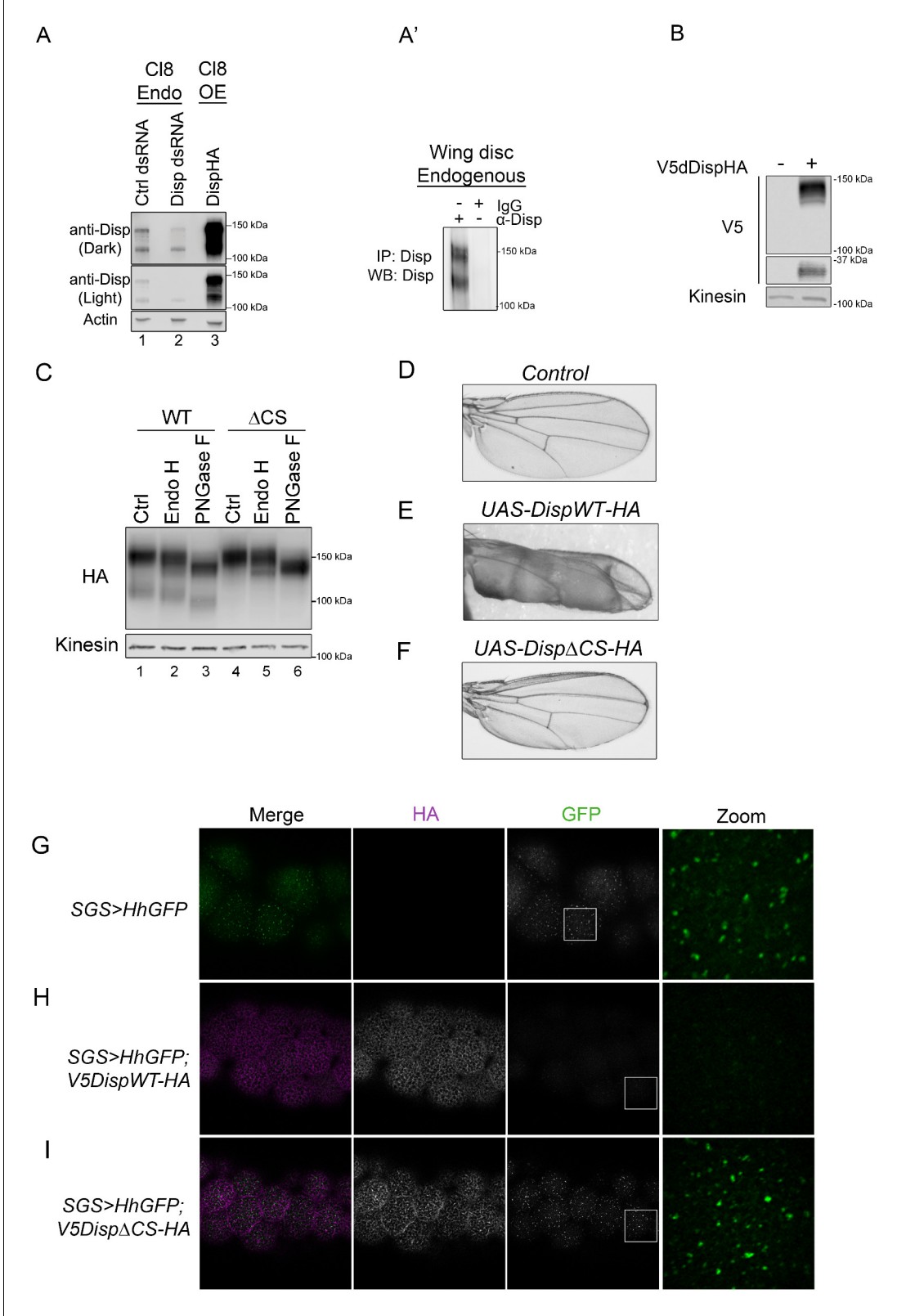

**Figure 5.** Cleavage is required for Disp activity in vivo. (**A**) Lysates from Drosophila Cl8 cells treated with control or *disp* dsRNA or transfected with *pAc-dispHA* were analyzed by western blot using anti-dDisp. Actin is the loading control. (**A'**) Endogenous dDisp150 and dDisp110 were specifically immunoprecipitated with anti-dDisp from wing imaginal disc lysate. (**B**) V5dDispHA was expressed in Cl8 cells and lysates were analyzed by western blot to confirm generation of the V5 fragment. Kinesin is the loading control. (**C**) Lysates were prepared from Cl8 cells expressing Δ206–238 (ΔCS)

*Figure 5 continued on next page*

*Figure 5 continued*

dDispHA protein and analyzed by western blot. Lysates were treated with Endo H or PNGase F. Kinesin is the loading control. (**D–F**) Wild type or ΔCS dDispHA proteins were expressed dorsally in wing imaginal discs using *apterous-GAL4*. Representative male wings are shown. (**G–I**) WT or ΔCS V5dDispHA proteins (magenta) were expressed with HhGFP (green) in salivary glands using *SGS-GAL4*. Maximum intensity projections of basolateral and basal optical sections of salivary glands are shown. Square in the GFP images indicates zoom area.

DOI: https://doi.org/10.7554/eLife.31678.013

The following figure supplement is available for figure 5:

**Figure supplement 1.**

DOI: https://doi.org/10.7554/eLife.31678.014

over-expression of dDispΔCS triggered modest wing curling, but did not induce pronounced blistering (*Figure 5F*). These results suggest compromised in vivo activity by the cleavage-deficient mutant.

To directly test for the effect of dDisp cleavage disruption on Hh export, a transgene encoding a Hh protein with an internal GFP that is retained post ligand processing (HhGFP [*Torroja et al., 2004*; *Hartman et al., 2013*]) was expressed alone or in combination with wild type or ΔCS V5dDispHA proteins in *Drosophila* salivary glands (*Figure 5G–I*). Salivary glands were chosen because they are large and do not express endogenous *disp* (modENCODE). These characteristics allowed for clear visualization of dDispHA effects on HhGFP without compensation by the endogenous protein. *UAS-hhGFP* and *UAS-V5dispHA* transgenes were recombined onto the same chromosome, and then expressed under control of *SGS-GAL4*. In control non-dDisp expressing salivary glands, Hh accumulated in large puncta on the basal surface of salivary gland cells (*Figure 5G*). Expression of wild-type V5dDispHA in salivary gland cells resulted in a mostly uniform membrane localization of HhGFP with distinct puncta evident throughout basolateral optical sections (*Figure 5H* and *Figure 5—figure supplement 1*). HhGFP was largely depleted at the basal surface but was evident in a small number of distinct puncta. Conversely, cells expressing V5dDispHAΔCS showed an overt increase in HhGFP puncta throughout basolateral optical sections with accumulation of large puncta at the basal surface that resembled puncta observed in the absence of V5dDispHA expression (*Figure 5I* and *Figure 5—figure supplement 1*). This punctate organization is similar to that observed for Hh protein expressed in embryonic and larval tissues (*Callejo et al., 2011*; *Gallet et al., 2003*), suggesting the puncta could represent 'packaged' HhGFP that has been primed for release. Accumulation of these puncta at the basal surface of DispΔCS-expressing salivary glands, where the majority of DispΔCSHA signal was detected, suggests that although HhGFP may appropriately package for release, it cannot effectively deploy when Disp cleavage is compromised. Thus, dDisp cleavage is required for ligand deployment in vivo.

Having established an evolutionarily conserved requirement for Disp processing for effective ligand release, we next wanted to examine the mechanism by which processing impacted Disp functionality. Disp is predicted to assemble into functional trimers (*Etheridge et al., 2010*), raising the possibility that cleavage might control oligomerization. To determine whether cleavage disruption attenuated Disp trimer assembly, wild type and CS murine DispHA proteins were expressed in NIH3T3 cells, and lysates were examined by native gel electrophoresis and western blot (*Figure 6A*). A ~ 480 kDa fraction consistent with the predicted molecular weight of the Disp trimer was evident for both wild type and cleavage deficient DispHA proteins, indicating that blocking cleavage did not block trimer formation (*Figure 6A*, top). Moreover, larger molecular weight fractions (~750 kDa) were present at equal intensities for both wild type and CS proteins, suggesting cleavage disruption did not prevent murine Disp from forming higher order assemblies.

Having confirmed DispCS was not deficient in trimer formation, we next assessed its ability to bind Shh. Disp is thought to bind Shh through a sterol sensing domain (SSD)-mediated association with the carboxyl-terminal Shh cholesterol modification, which is subsequently transferred to Scube2 (*Tukachinsky et al., 2012*). Disp EC1, which contains the processing site, is situated directly adjacent to the SSD (*Figure 2B*). As such, Disp cleavage could potentially influence Shh association by governing SSD access. To test this, wild type and CS mutant V5DispHA proteins were co-expressed with Shh in *Disp-/-* cells, and the ability of Shh to co-immunoprecipitate with DispHA from cellular lysates was examined. Similar amounts of Shh co-immunoprecipitated on anti-HA beads with both WT and CS mutant DispHA proteins (*Figure 6B*, lanes 3 and 4). Shh-DispHA binding was specific because

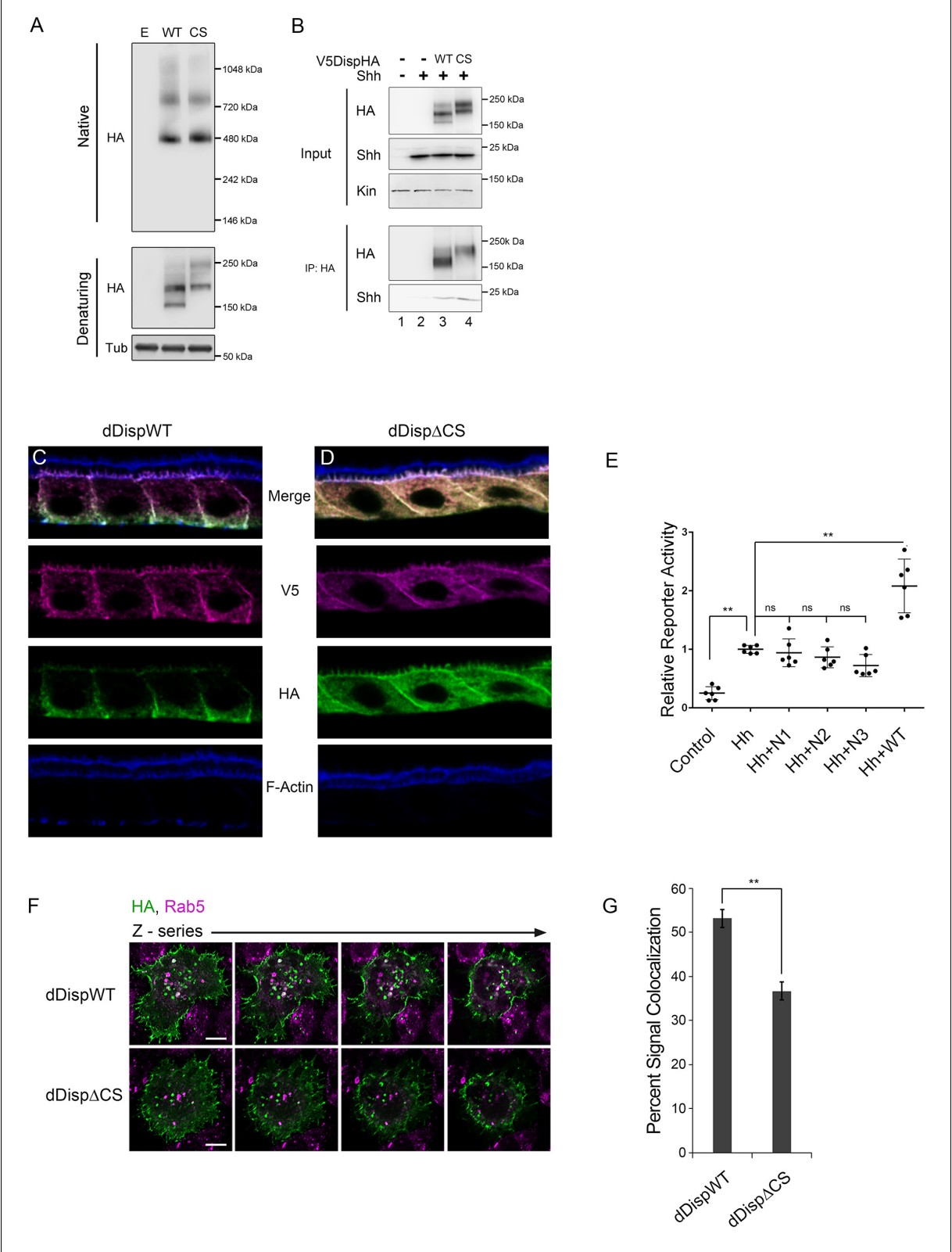

**Figure 6.** Processing impacts Disp membrane localization. (**A**) Lysates from NIH3T3 cells expressing wild type or CS mutant Disp proteins were analyzed by native (upper) and denaturing (lower) gel electrophoresis and western blot. The predicted molecular weight of the Disp trimer is ~480 kDa. Tubulin is the loading control. (**B**) DispHA proteins were immunoprecipitated with anti-HA antibody from lysates of *Disp-/-* cells expressing Shh alone or with wild type or CS DispHA. Wild type and CS mutant DispHA proteins co-immunoprecipitate Shh (bottom). (**C–D**) Wild type and ΔCS V5dDispHA

*Figure 6 continued on next page*

Figure 6 continued

proteins were expressed in ovarian follicle cells using the *C204-GAL4*. F-actin marks apical membrane in follicle cells (blue). The dDisp amino-terminal region is indicated by V5 (magenta) and carboxyl domain by HA (green). (**E**) V5dDispHA or increasing amounts of the amino-terminal V5 fragment were expressed with Hh in ligand producing cells, and then co-cultured with Cl8 cells transfected with Hh-responsive *ptcΔ136-luciferase* reporter and *actin-renilla* control. Luciferase reporter activity in ligand receiving cells was measured, and is shown normalized to renilla and relative to the Hh response in reporter cells co-cultured with empty vector control transfected cells. The experiment was repeated three times in duplicate and all data pooled. Error bars indicate s. d. Significance was determined by a one-way ANOVA. (**F–G**) S2 cells expressing wild type or ΔCS V5dDispHA proteins were analyzed for colocalization (white) between HA (green) and endogenous Rab5 signals (magenta) by confocal microscopy. Serial z-sections are shown. Scale bar is 5 μm. Percent colocalization between HA and Rab5 signals was determine by Imaris image analysis software (**H**). Error bars indicate s. e. For all panels, **p≤0.01; ns, not significant.

DOI: https://doi.org/10.7554/eLife.31678.015

The following source data is available for figure 6:

**Source data 1.** Data for *Figure 6E*.

DOI: https://doi.org/10.7554/eLife.31678.016

Shh failed to bind HA beads in the absence of DispHA (lane 2). These results suggest that Disp cleavage does not regulate ligand binding.

Disp predominately localizes to basolateral membranes in polarized epithelial cells, but a minor sub-apical, vesicular pool has been reported (*Etheridge et al., 2010*; *Callejo et al., 2011*). Cholesterol-modified Hh ligand enriches apically, placing the majority of Disp protein and its target ligand in non-overlapping membrane domains (*D'Angelo et al., 2015*; *Callejo et al., 2011*). In *Drosophila*, dDisp achieves ligand release by capturing apical Hh in recycling endosomes, which subsequently retarget to the plasma membrane for ligand deployment (*D'Angelo et al., 2015*; *Callejo et al., 2011*). Because convertase-mediated cleavage can impact protein function by affecting intracellular trafficking (*Constam, 2014*), we hypothesized Disp cleavage might affect its membrane targeting. To assess its subcellular localization in polarized cells in vivo, we tested V5dDispHA subcellular localization in ovarian follicle cells, which are large and polarized, making them ideal for monitoring protein trafficking and subcellular localization. Wild type and ΔCS V5dDispHA proteins were expressed using the follicle cell driver *C204-GAL4*, and localization of V5 (amino) and HA (carboxyl) epitope tags was examined in stage 10 ovaries (*Figure 6C–D*). Colocalization between amino-V5 (magenta) and carboxyl-HA (green), indicative of the unprocessed protein, was evident for wild type V5dDispHA in basal vesicles and along basolateral membranes (*Figure 6C*, white; F-Actin marks apical, blue). Consistent with processing removing the amino-terminal fragment from dDisp110, a clear separation of the two signals was observed. The released amino-terminal V5 fragment, evidenced by V5 signal not colocalized with HA, localized to apical and basolateral membrane and in vesicles throughout the cell (*Figure 6C*, magenta). Notably, dDisp110HA depleted apically, enriching on basolateral membrane and in basally-localized vesicles (*Figure 6C*, green), suggesting differential trafficking of the two dDisp domains post-cleavage. Although we cannot rule out a functional role for the amino-terminal fragment post-cleavage, we do not think it contributes to signaling because its over-expression in wing disc-derived Cl8 cells did not alter induction of Hh-dependent *luciferase* reporter gene activity (*Figure 6E*). We were unable to directly confirm activity of DispΔN lacking the amino-terminal prodomain due to it being retained in the ER (not shown). However, the functional pool of Disp protein is thought to enrich basolaterally in both murine and *Drosophila* systems, which is consistent with what we observed for dDisp110HA in follicle cells (*Etheridge et al., 2010*; *Callejo et al., 2011*).

Visualization of cleavage-deficient V5dDispΔCSHA revealed a strikingly altered localization from that of the wild-type protein (*Figure 6D*). The cleavage site mutant showed pronounced accumulation on both apical and basolateral membranes, along with uniform vesicular distribution throughout the cell, suggestive of altered membrane trafficking upon cleavage loss. Notably, signal intensity of cleavage-deficient dDisp was increased compared to wild type, potentially indicating that dDisp protein turnover might be affected by altered membrane recycling.

To directly test whether dDisp membrane recycling was altered by cleavage disruption, we expressed wild type and ΔCS V5dDispHA proteins in S2 cells, and tested for dDisp colocalization with the early endosomal marker Rab5 by immunofluorescence confocal microscopy (*Figure 6F*, white). Imaris image analysis software was used to perform colocalization analysis of ~50

V5dDispHA-expressing cells per condition across three independent experiments. In cells expressing wild-type V5dDispHA, approximately 50% of the HA signal was colocalized with endogenous Rab5 signal. Cleavage site deletion lessened V5dDispHA-Rab5 colocalization, reducing the percent of HA signal colocalized with Rab5 signal to ~35% (*Figure 6G*). These results are consistent with compromised membrane recycling, and taken together with in vivo experiments, suggest that Disp cleavage is necessary for proper membrane trafficking.

## Discussion

Disp was first identified as a crucial regulator of Hh ligand deployment in 1999 through a genetic screen conducted in *Drosophila* (*Burke et al., 1999*). A number of vertebrate genetic studies subsequently established the importance of Disp in Shh morphogen gradient formation and activity during tissue development (*Caspary et al., 2002*; *Ma et al., 2002*; *Kawakami et al., 2002*; *Nakano et al., 2004*). Owing to the comparatively small number of cell biological and biochemical interrogations of Disp activity (*Creanga et al., 2012*; *Tukachinsky et al., 2012*; *Etheridge et al., 2010*), mechanistic insight into its regulation and functionality has remained limited. The study presented here improves understanding of Disp regulation by revealing an evolutionarily conserved cleavage event that influences the ability of Disp to deploy Hh family ligands from ligand-producing cells. We report that Disp is cleaved at a conserved processing site in its predicted first extracellular loop by the proprotein convertase Furin. Cleavage site mutation compromises Disp-mediated ligand deployment in vitro and in vivo, leading to reduced pathway activation in target cells. As such, this study is the first to provide mechanistic insight into a process promoting functional maturation of Disp for its role in ligand deployment.

Proprotein convertase-mediated cleavage of substrate proteins typically promotes their maturation by removing inhibitory prodomains, revealing active domains, releasing bioactive fragments, priming substrates for cleavage by additional proteases, or by influencing substrate subcellular localization (*Seidah et al., 2013*; *Seidah and Prat, 2012*). How prodomains affect substrate trafficking is not yet fully understood, but a logical hypothesis is that cleavage regulates association with trafficking molecules and/or tethering proteins along secretory or endosomal recycling routes. Such a model has been proposed for the convertase substrate Nodal, which accumulates on the cell surface following processing inhibition (*Constam, 2014*; *Blanchet et al., 2008a*, *2008b*). Because cleavage disruption altered full-length dDisp membrane localization in polarized epithelial cells to mimic what was observed for the processed 30 kDa V5 fragment, we hypothesize that like Nodal, Disp membrane trafficking is regulated in cleavage-dependent manner. The observed colocalization of amino- and carboxyl-terminal dDispWT epitope tags on basolateral membranes of *Drosophila* follicle cells suggests cleavage occurs after basolateral targeting of unprocessed Disp. Consistent with this hypothesis, Furin has been demonstrated to traffic basolaterally in polarized epithelial cells (*Simmen et al., 1999*).

Intriguingly, whereas Disp is predominantly observed to localize to basolateral membrane, Hh enriches on apical membrane, from which it must be endocytosed in a Disp-dependent manner to facilitate its release upon plasma membrane recycling (*D'Angelo et al., 2015*; *Callejo et al., 2011*; *Gallet et al., 2003*). Our observations that cleavage-deficient Disp (1) accumulated uniformly along apical and basolateral membranes of *Drosophila* follicle cells and (2) showed reduced colocalization with the Rab5 endosomal marker when expressed in S2 cells suggest that its endosomal trafficking is likely compromised by cleavage disruption. As such, we suggest a testable model in which basolateral Disp cleavage activates the protein for endosomal recycling, allowing it to capture, recycle and deploy apically localized Hh. We do not believe Disp cleavage is required to interact with ligand because both wild type and cleavage-deficient murine Disp proteins co-immunoprecipitated with Shh. Cleavage-deficient Disp was also capable of forming multimers, diminishing the likelihood that EC1 clipping regulates self-association.

Results obtained using both murine and *Drosophila* experimental systems demonstrate that Disp processing is evolutionarily conserved. However, whereas mutation of a single consensus cleavage motif in EC1 of mouse Disp was sufficient to disrupt cleavage, multiple predicted sites had to be targeted in *Drosophila* Disp EC1. That three predicted motifs had to be deleted to block *Drosophila* Disp cleavage suggests cleavage site redundancy in the fly protein. Multiple redundant motifs may indicate increased reliance upon Disp cleavage for function in the *Drosophila* system. Notably,

*Drosophila* lack a Scube2-like protein that partners with dDisp to extract Hh from ligand-producing cells (*Creanga et al., 2012*; *Tukachinsky et al., 2012*). Multiple redundant sites might serve as fail-safes to assure dDisp cleavage and efficient Hh membrane release in the absence of Scube2-mediated assistance. Another possible explanation is that *Drosophila* Disp is cleaved by additional or alternative proteases with different cleavage site preferences or efficiencies. It has been reported that although *Drosophila* convertases can share substrate specificity with their vertebrate counterparts, cleavage efficiency will often vary between the two systems (*De Bie et al., 1995*).

In vertebrates, genetic loss-of-function of proprotein convertases such as Furin, PCSK5 and PACE4 triggers developmental defects leading to embryonic lethality (*Seidah et al., 2013*; *Roebroek et al., 1998*; *Essalmani et al., 2008*; *Constam and Robertson, 2000a*; *Constam and Robertson, 2000b*). Although *Furin* and *Disp1* knockout mice both show axial rotation and heart looping defects that lead to death at or before embryonic days ~ E9.5-10.5, their phenotypes are not indistinguishable (*Caspary et al., 2002*; *Ma et al., 2002*; *Kawakami et al., 2002*; *Roebroek et al., 1998*). Most notably, whereas *Disp1* mutant embryos show clear disruption of left-right asymmetry, *Furin* mutants do not (*Ma et al., 2002*; *Kawakami et al., 2002*; *Roebroek et al., 1998*). This could be due to functional compensation by other convertases in vivo. Consistent with this notion, functional redundancy between Furin, PACE4, PCSK5 and PCSK7 has been reported (*Seidah et al., 2013*; *Roebroek et al., 1998*). It is also possible that Furin-mediated Disp cleavage occurs in temporal or tissue-specific manners to scale Shh release efficiency commensurate with increased need. In such a scenario, Disp cleavage would be predicted to occur during later developmental stages or in larger developing tissues to bolster Shh deployment for a growing population of target cells. Future in vivo studies using vertebrate model systems will be required to explore these hypotheses, and to determine how Disp cleavage disruption impacts Shh-dependent developmental patterning.

## Materials and methods

### Experimental procedures

**Key resources table**

| Reagent type (species) or resource | Designation | Source or reference | Identifiers | Additional information |
|---|---|---|---|---|
| Genetic reagent (*Drosophila melanogaster*) | Oregon-R | Bloomington Stock Center, Bloomington, IN | | |
| Genetic reagent (*D. melanogaster*) | apterous-Gal4 | Bloomington Stock Center | | |
| Genetic reagent (*D. melanogaster*) | SGS-Gal4 | Bloomington Stock Center | | |
| Genetic reagent (*D. melanogaster*) | UAS-HhGFP | *Hartman et al. (2013)*, 201(5) 741–57. | | |
| Genetic reagent (*D. melanogaster*) | UAS-dispWT | This study | | |
| Genetic reagent (*D. melanogaster*) | UAS-dispΔCS | This study | | |
| Cell line (*Mus musculus*) | NIH3T3 | ATCC, Manassas, VA | CRL-1658 | |
| Cell line (*Homo sapiens*) | HEK293T | ATCC | CRL-11268 | |
| Cell line (*M. musculus*) | Dispatched KO MEFs | *Ma et al. (2002)* 111(1): 63–75 | | |
| Cell line (*H. sapiens*) | LoVo | ATCC | CCL-229 | |
| Cell line (*H. sapiens*) | HCT-15 | ATCC | CCL-225 | |

*Continued on next page*

*Continued*

| Reagent type (species) or resource | Designation | Source or reference | Identifiers | Additional information |
|---|---|---|---|---|
| Cell line (*M. musculus*) | Light II | ATCC | JHU-68 | |
| Cell line (*D. melanogaster*) | Cl8 | DGRC, Bloomington, IN | stock # 151 | |
| Cell line (*D. melanogaster*) | S2 | Thermo Fisher, Waltham, MA | R69007 | |
| Cell line (*M. musculus*) | Furin-/- | This paper | CRISPR/Cas9 | C57BL/6 MEF cells |
| Cell line (*M. musculus*) | Pcsk5-/- | This paper | CRISPR/Cas9 | C57BL/6 MEF cells |
| Cell line (*M. musculus*) | Pace4-/- | This paper | CRISPR/Cas9 | C57BL/6 MEF cells |
| Cell line (*M. musculus*) | Pcsk7-/- | This paper | CRISPR/Cas9 | C57BL/6 MEF cells |
| Antibody | anti-Kif5b | Abcam, Cambridge, MA | ab167429 | 1:5000 (WB) |
| Antibody | anti-tubulin | Cell Signaling, Danvers, MA | 3873 | 1:10000 (WB) |
| Antibody | anti-Myc Affinity gel | Sigma, St Louis, MO | E6654 | 25 ul slurry for IP |
| Antibody | anti-HA affinity gel | Sigma | E6779 | 25 ul slurry for IP |
| Antibody | anti-Shh | Santa Cruz Biotechnolgy, Dallas, TX | sc-9024 | 1:2000 (WB) |
| Antibody | anti-Myc | Roche, Basal, Switzerland | 11667149001 | 1:1000 (WB) |
| Antibody | anti-Disp | This Study | | 1:1000 |
| Antibody | anti-mouse HRP | Jackson Immuno, West Grove, PA | 715-035-151 | 1:10000 |
| Antibody | anti-Rabbit HRP | Jackson Immuno | 711-035-152 | 1:10000 |
| Antibody | anti-Rat HRP | Jackson Immuno | 112-035-175 | 1:10000 |
| Antibody | AlexaFluor 488 | Life technologies, Carlsbad, CA | A11029 (Mouse) A11034 (Rabbit) A11006(Rat) | 1:1000 |
| Antibody | AlexaFluor 555 | Life technologies | A21424 (Mouse) A21429 (Rabbit) A21434(Rat) | 1:1000 |
| Antibody | AlexaFluor 633 | Life technologies | A21236 (Mouse) A21245 (Rabbit) A21247(Rat) | 1:1000 |
| Antibody | anti-Mouse IR800 | LiCor, Lincoln, NE | 92632212 | 1:10000 |
| Antibody | anti-Rabbit IR800 | LiCor | 92632213 | 1:10000 |
| Antibody | anti-Rat IR800 | LiCor | 92632219 | 1:10000 |
| Antibody | | | | |
| Transfected construct (*M. musculus*) | Dispatched | This paper | RIKEN | http://dna.brc.riken.jp/ |
| Transfected construct (*M. musculus*) | pCDNA3 Dispatched WT HA | This paper | | pCDNA3 from Invitrogen |
| Transfected construct (*M. musculus*) | pCDNA3 V5 Dispatched WT HA | This paper | | V5 introduced following A106 |
| Transfected construct (*M. musculus*) | pCDNA3 V5 Dispatched CS HA | This paper | | Mutate R279, E280 to alanine |
| Transfected construct (*M. musculus*) | MSCV Hygro Shh-FL | This paper | | MSCV Hygro from Clontech |
| Transfected construct (*M. musculus*) | pCDNA3-V5 Disp TM4/TM10HA | This paper | | |

*Continued on next page*

*Continued*

| Reagent type (species) or resource | Designation | Source or reference | Identifiers | Additional information |
|---|---|---|---|---|
| Transfected construct (*M. musculus*) | pFLC-I-Scube2 | SourceBioscience, Nottingham, UK | Clone E030016G24 | |
| Transfected construct (*M. musculus*) | pCDNA3-Scube2 Flag | This paper | | |
| Transfected construct (*H. sapiens*) | pCMV6 - huFurin Myc DDK | This paper | Origene, RC204279 | |
| Transfected construct (*M. musculus*) | pCDNA3 - Shh-FL | This paper | | Gift from P. Beachy Lab |
| Transfected construct (*Aequorea victoria*) | pCDNA3 - GFP | This paper | | |
| Transfected construct (*D. melanogaster*) | pFLC-I-disp cDNA | This paper | DGRC | Supported by NIH grant 2P40OD010949 |
| Transfected construct (*D. melanogaster*) | pAc-dispHA | This paper | | Cloned into pAc5.1 Vector with HA tag |
| Transfected construct (*D. melanogaster*) | pAc-V5dispHA | This paper | | V5 introduced following V108 |
| Transfected construct (*D. melanogaster*) | pAc-disp Δ 206–238 HA | This paper | | |
| Transfected construct (*D. melanogaster*) | pUAS-aatB-V5dispWTHA | This paper | | |
| Transfected construct (*D. melanogaster*) | pUAS-aatB-V5disp Δ206–238 HA | This paper | | |
| Transfected construct (*D. melanogaster*) | pUAS-HhGFP | This paper | | |
| Transfected construct (*Photinus pyralis*) | ptcΔ 136-luciferase | This paper | | |
| Transfected construct (*Renilla reniformis*) | pAc-renilla | This paper | | |
| Transfected construct (*D. melanogaster*) | pAC-hh | This paper | | |
| Software, algorithm | Adobe Photoshop CS4 | Adobe, San Jose, CA | | for making figures |
| Software, algorithm | LAS X | Leica, Wetzlar, Germany | | image analysis |
| Software, algorithm | Prism | GraphPad, La Jolla, CA | | for statistical analysys and graphs |
| Software, algorithm | Huygens Professional software | | | for decovolution images |
| Commercial assay or kit | Quickchange II XL Kit | Agilent, Santa Clara, CA | 200522 | |
| Commercial assay or kit | Lipofectamine 2000 | ThermoFisher Scientific | 11668027 | |
| Commercial assay or kit | Lipofectamine 3000 | ThermoFisher Scientific | L3000008 | |
| Commercial assay or kit | FuGene HD | ThermoFisher Scientific | PRE2311 | |
| Commercial assay or kit | ECL Prime Western Blotting Detection Reagent | Fisher Scientific, Hampton, NH | RPN2232 | |
| Commercial assay or kit | Dual Luciferase Reporter Assay Kit | Promega, Madison, WI | PRE1960 | |
| Chemical compound, drug | Furin I Inhibitor | Enzo Life Sciences, Farmingdale, NY | ALX-260–022 M005 | |
| Chemical compound, drug | MG-132 | EMD Chemicals Inc., St. Louis, MO | 474790 | |

## Cell lines

NIH3T3 (CRL-1658), HEK293T (CRL-11268), LoVo (CCL-229), HCT-15 (CCL-225) and LightII (JHU-68) cells were obtained from ATCC, S2 cells from ThermoFisher (R690-07), and Cl8 cells (CME W1 Cl.8 +) were obtained from DGRC. *Disp-/-* knockout MEFs were obtained from P. Beachy and A. Salic (*Ma et al., 2002*; *Tukachinsky et al., 2012*).

*Furin-/-*, *Pcsk5-/-*, *Pace4-/-*, and *Pcsk7-/-* cell lines were generated using CRISPR/Cas9 technology. C57BL/6 MEF cells were transiently transfected with 3.5 μl of Cas9 RNP (Cas9 (Berkeley Macrolab), 40 pmole; sgRNA (Synthego, Redwood City, CA),156 pmole) via nucleofection (Lonza, 4D-Nucleofector™ X-unit, Basal, Switzerland) using solution P3, program DS-150 in small cuvettes according to the manufacturers recommended protocol. sgRNAs used were: Furin: 5'- TCTGTAGCCGGCTG TGCCGC; Pcsk5: TGGAAAGAAACCTTGGTACT; Pace4: TACCACATGTTAGACCAAAT; Pcsk7: TTG TGGTTGCCAGTGGTAAT. Cells were single-cell sorted by flow cytometry 3 days post-nucleofection, clonally expanded and verified for disruption of the endogenous locus via western blot for protein expression if antibodies were available, and/or targeted deep sequencing to identify frameshift mutations.

All cell lines were routinely validated by functional assay and western blot as appropriate, and screened monthly for mycoplasma contamination by PCR. Commercially available cell lines are re-ordered quarterly. Cells were cultured as described below.

Plasmids, transgenes, *Drosophila* embryo injection, protein expression and antibody generation *pCDNA3-Disp* was generated by introducing *Disp1* cDNA from RIKEN (Wako, Japan, (http://dna.brc.riken.jp/)) into Not1-Xba1 sites in *pCDNA3* (Invitrogen). The HA tag was introduced as an annealed oligo into the Xho1-Xba1 site in *pCDNA3* using primers (forward 5' tcgagtacccctac-gatgtgcccgattatgcatacccatacgatgttccagattacgctgtttaat and reverse 5' ctagattaaacagcgtaatctggaacatcgtatgggtatgcataatcgggcacatcgtaggggtac). pCDNA3-Scube2Fg was generated from pFLC-I-Scube2 (SourceBiosciences, E30016G2) by sub-cloning into the Not1/Xho1 site.

To generate double-tagged Disp, V5 epitope tag coding sequence was introduced behind amino acid Alanine106 of pCDNA3-DispHA using the primers (forward 5' gaggctggccttgcaggtaagcctatccc-taaccctctcctcggtctcgattctacggcctcccccgctttg and reverse 5' caaagcggggggaggccgtagaatcgagaccgag-gagagggttagggataggcttacctgcaaggccagcctc). Mutagenesis of the cleavage site was performed using the Quickchange II XL kit (Agilent) using forward 5' gatcaccatgagagagagagaGCAGCAgtg-gactggaacttccagaaag and reverse primer 5' ctttctggaagttccagtccactgctgctctctctctcatggtgatc.

*Drosophila disp* was amplified from *pFLC-I-disp* cDNA (DGRC), and inserted in frame with an HA epitope tag into the *pAc5.1* vector (Invitrogen) to generate *pAc-dispHA*. The cleavage site deletion (Δ206–238) mutant was generated by Quickchange mutagenesis (Agilent) of *pAc-dispHA*. To make the double tagged construct, sequence encoding the V5 epitope tag was introduced following V108 to generate *pAc-V5dispHA*. To generate transgenic Drosophila, *V5dispHA* and *V5dispHAΔCS* were sub-cloned from *pAc5.1* into *pUAS-attB* (*Bischof et al., 2007*). Transgenes were targeted to landing site 68E1 on chromosome 2. Embryo injections were performed by Best Gene, Inc. For salivary gland analysis, *UAS-HhGFP* and *UAS-V5dispHA* transgenes were recombined using standard methods.

To generate antisera against *Drosophila* Disp, the coding region of the predicted fourth extracellular loop (amino acids: 694–959) was introduced into *pET-28b* in frame with a carboxyl terminal 6X His tag. Protein was expressed in BL-21 cells and affinity purified on nickel resin by standard methods. Antisera were produced in rabbits using the Covance custom antibody service.

## Cell transfection

For insect cell transfections, approximately $3 \times 10^6$ Clone 8 (Cl8) cells were plated in M3 insect media (Sigma) plus 10% fetal bovine serum (FBS) and 2% fly extract in 60 mm dishes the day before transfection. The following morning, cells were transfected with 2 μg of *pAc5.1* expression vectors for Disp or Hh proteins using Lipofectamine 2000 (Invitrogen). DNA content was normalized with empty *pAc5.1* vector.

For mammalian cell transfection, HEK293T, NIH3T3, LoVo, HCT-15, *Furin-/-* or *Disp-/-* cells were seeded at a density of $1 \times 10^6$ cells/60 mm dish in DMEM plus 10% bovine calf serum or DMEM plus 10% FBS form MEFs. Empty *pCDNA3* (2 μg), *pCDNA3-DispHA* (2 μg), *pCMV6-huFurin* (1 μg,

Origene), *pCDNA3-Shh* (1 µg) and/or *pCDNA3-GFP* (1 µg) constructs were transfected into NIH3T3, LoVo, HCT-15, *Furin-/-* or *Disp-/-* cells using Lipofectamine 2000 or 3000 (Invitrogen).

## Immunofluorescence microscopy

For immunofluorescence analysis of *Drosophila* ovaries, tissue was dissected from 2 to 3 day old *C204 >V5 dispHA* (WT or ΔCS) females using standard methods. Samples were imaged on a Leica TCS SP8 confocal microscope with a 1.4NA 63X objective and 0.7 AU pinhole using spatial sampling matching nyquest criteria. Images were deconvolved using Huygens Professional software (theoretical PSF, Classic Maximum Likelihood Estimation (CMLE) algorithm, with five iterations, max) and processed using LAS X and Adobe Photoshop CS4. V5 and HA epitope tags were detected using Anti-V5 (1:500; Life Technologies) along with AlexaFluor 488 (1:1000; Life Technologies) and anti-HA (1:250; Roche) along with AlexaFluor 555 (1:1000; Life Technologies) respectively. Rab5 was detected using anti-Rab5 (1:100, Abcam) and AlexaFluor secondary antibody (1:1000, Life Technologies). Phalloidin conjugated with AlexaFluor 633 (1:100; Life Technologies) was used to mark F-actin.

For Rab5 colocalization analysis, confocal images were acquired in z-stacks (3–5 slices with a slice interval of 0.25 µm) using a Zeiss LSM780 microscope. Colocalization analysis was done using Imaris image analysis software. The 'Spots' function was used to define HA and Rab5 puncta on all slices of each cell. The 'Colocalize Spots' function was used to identify the number of HA spots colocalized with Rab5 spots, using preset Imaris parameters. Fifty V5dDispHA-expressing cells per condition were selected at random, and analyzed over three independent experiments to determine percent signal colocalization. Significance was determined using Student's t-test.

## Functional assays, cell lysis and western blots

For dDisp expression analyses in insect cells, membrane fractions were isolated from Cl8 cells in modified HK Buffer (HK Buffer (20 mM Hepes, 10 mM KCl; pH 7.9)+5% Glycerol+150 mM NaCl) as described (*Ogden et al., 2003*).

For Shh release and Disp expression analyses in *Disp-/-* or *Furin-/-* cells, transfected cells were washed twice in serum-free DMEM, then incubated for 6 hr in serum-free DMEM with three media changes during incubation. Shh conditioned media was collected by incubating washed cells in 2 mL serum-free DMEM for ~48 hr. Conditioned media was centrifuged at 4°C for 1 hr at 9000 x g. The resulting supernatant was centrifuged an additional hour at 16,000 x g. Supernatant was TCA precipitated for six hours at 4°C before pelleting and re-suspending in TCA Resuspension Buffer (2% w/v SDS, 0.42M Tris-HCl, pH 7.4, 4% v/v glycerol, 0.01% w/v Bromphenol Blue and 0.05M DTT) as described (*Goetz et al., 2006*). Protein concentration was determined by BCA assay (Pierce, Waltham, MA) and equal total protein amounts for each sample analyzed by SDS-PAGE on Criterion gels (Biorad, Hercules, CA) and western blot.

For NIH3T3 cell lysis, cells were washed twice in 1X PBS, harvested in 1% NP-40 Lysis Buffer (50 mM Tris-HCl, pH 8.0, 150 mM NaCl, 1% NP-40, 0.1% SDS, 1X Protease Inhibitor Cocktail and 0.5 mM DTT) and incubated for 30 min at 4°C. Extracts were cleared by centrifugation at 14,000 x g at 4°C for 45 min and analyzed as above.

For western blotting, SDS-PAGE samples were transferred onto Protran Nitrocellulose (GE) or Immobilon-P PVDF (Millipore) using Tris/Glycine/SDS Buffer (Biorad) at 100V for one hour at 22°C. Membranes were blocked with 5% milk and 0.1% Tween-20 in Tris-buffered saline (TBS) for 1 hr at room temperature. Nitrocellulose membranes were immunoblotted for 1 hr at 22°C using anti-HA (1:5000; Covance, Princeton, NJ or 1:3000; Roche), anti-V5 (1:5000; Life Technologies), anti-Hh (1:1000; SCBT), Drosophila Kinesin (1:10,0000; Cytoskeleton Inc., Denver, CO), Actin (1:10,000; Millipore), Mouse Kinesin (anti-Kif5B, 1:5000; Abcam), and/or Tubulin (1:10,000; Cell Signaling) followed by three 5-min washes in secondary milk (primary milk diluted to 25% with TBS). Corresponding HRP-conjugated secondary antibodies (Jackson Immuno, West Grove, PA) were incubated for 1 hr at RT at a 1:10,000 concentration. Infrared antibodies (Li-Cor) were used at a 1:10,000 concentration with HRP-conjugated antibodies when duplexing. Blots were developed on film or by using an Odyssey Fc (Li-Cor) with ECL Prime (GE, Pittsburgh, PA).

For murine co-culture reporter assays, *Disp-/-* cells stabling expressing *MSCV-Hygro* or *MSCV Hygro-Shh* were seeded at a density of $1 \times 10^6$ cells per 60 mm plate in DMEM-10% Fetal Bovine Serum complete media. The following day, *pCDNA3-GFP* (2 µg), *pCDNA3-DispHA* (2 µg), *pCDNA3-*

DispCSHA (2 μg) and pCDNA3-DispTM4/TM10HA (2 μg) were transfected into Disp-/- cells expressing vector or Shh using Lipofectamine 3000. The following day, FlexiPerm discs (Sarstedt, Germany) were sterilized in 70% ethanol, dried and placed in the center of each well in a 6-well dish, creating a barrier between the inner ring of the well and the outer ring of the well. LightII reporter cells were seeded at $1 \times 10^6$ cells per well on the outer ring of the FlexiPerm disc in DMEM-10% BCS complete media. Disp-/- stable cells transfected with the indicated Disp expression vectors were seeded at a density of $1 \times 10^5$ cells per well in the inner ring of the FlexiPerm disc. Cells were allowed to recover for 4 hr. Media was removed from the cells and the FlexiPerm discs were removed creating a cell-free barrier between the LightII and Disp-/- cells. Cells were washed with PBS and then DMEM Serum Free Complete media. DMEM Serum-Free Complete media was added back to each well and allowed to incubate for 2 hr. Washing was carried out over 6 hr repeating the above wash steps. After 6 hr, 3mls of DMEM Serum Free Complete Media was added to each well and the cells were allowed to incubate for ~36 hr. Reporter assay were carried out according to Dual Luciferase Reporter Assay Kit instructions (Promega). Experiments were repeated four times in triplicate or quadruplicate, and all data pooled. Error bars indicate s.e.m. Significance was determined using a one-way ANOVA.

For insect cell reporter assays, Cl8 cells were plated in 60 mm culture dishes the day before transfection and grown to ~70% confluency and transfected using Lipofectamine 2000. Twenty-four hours post-transfection, reporter cells transfected with ptcΔ136-luciferase (600 ng) reporter construct and pAc-renilla (60 ng) normalization control were combined in a 1:3 ratio with ligand-producing cells transfected with pAc-hh (1 μg) and wild type or V5 fragment pAc-disp (1X = 500 ng) in 12-well culture dishes. Cells were co-cultured for ~48 hr and processed using the Dual Luciferase kit (Promega).

## Shh and Furin immunoprecipitation

Proteins of interest were expressed in NIH3T3 or HEK293T cells. Cell lysates were prepared ~48 hr post-transfection using RIPA lysis buffer (Millipore, Burlington, MA). Co-immunoprecipitation assays were performed as described (Marada et al., 2015) with the following modifications. EZview Red Anti-HA Affinity Gel (Sigma) and EZview Red Anti-c-Myc Affinity Gel (Sigma) were used to immunoprecipitate HA and Myc epitope-tagged proteins respectively. Immunoprecipitates were analyzed by western blot using the following antibodies: Anti-HA (1:2000, Roche), anti-Shh (1:2000, SCBT), anti-Kin/mKif5B (1:10,000, Cell Signaling), anti-Myc (1:1000, Roche), anti-Flag (1:2000, Sigma), anti-Furin (1:1000, SCBT) and anti-Tub (1:10,000, Cell Signaling).

## Drosophila Disp immunoprecipitation from wing discs

Wing imaginal discs from six to eight wild type (Oregon R) third instar larva were homogenized in RIPA lysis buffer (0.05M Tris-HCl, pH 7.4, 0.15M NaCl, 0.25% deoxycholic acid, 1% NP-40, 1 mM EDTA and 0.1% SDS). Lysates were centrifuged for 10 min at 2000 x g and supernatant was pre-cleared with 30 μL of a 50% A/G plus agarose slurry for 30 min. Supernatants were incubated with 10 μg anti-Disp antibody or rabbit IgG for 2 hr with gentle rocking at 4°C. Immune complexes were collected on 30 μL of A/G bead slurry for 60 min at 4°C. Beads were washed twice in lysis buffer and associated proteins were eluted by boiling for 5 min in 2x sample buffer (2% w/v SDS, 2 mM DTT, 4% v/v glycerol, 0.04 M Tris-HCL, pH 6.8% and 0.01% w/v Bromphenol blue) and analyzed by SDS-PAGE and western blot.

## Deglycosylation analysis and cell surface biotinylation

Deglycosylation, biotinylation and densitometry analysis were preformed exactly as previously described (Marada et al., 2013, 2015).

## Drug treatment

Transfected cells were treated with MG132 (50 μM), or Furin Inhibitor I (10, 25, 50, 75 and 100 mM) in serum-free DMEM for ~6 (MG132) or ~8 (Furin Inhibitor I) hours prior to cell lysis.

## Native gel electrophoresis

NIH3T3 cells were transfected with pCDNA-DispHA, pCDNA-DispCSHA or empty vector control. Lysates were processed as previously described (Etheridge et al., 2010) with slight modifications.

Approximately 48 hr post transfection cells were lysed for 30 min on ice in 1x NativePAGE sample buffer containing protease inhibitor cocktail (Roche) and 1% n-dodecyl-B-d-maltoside (DDM). Lysates were treated with Benzonase nuclease (Sigma) for 30 min at room temperature followed by centrifugation at 12,000xg for 30 min at 4°C. Supernatants were collected and run on a 4–20% NativePAGE Bis-Tris gel and transferred to PVDF membrane. A fraction of the lysates was run on a 7.5% Tris-HCl gel (BioRad) using denaturing settings and transferred to nitrocellulose membrane. Blots were probed using anti-HA antibody (Roche) and protein sizes were determined using NativeMark on native gels and Precision plus protein standard (Biorad) on denaturing gels. NativePAGE sample buffer, DDM, NativePAGE gels and NativeMark molecular weight standard were purchased from Life Technologies.

### Edman sequencing sample preparation

HEK293T cells were seeded in thirty 100 mm plates at a density of $8 \times 10^6$ cells/plate in DMEM plus 10% Fetal Bovine Serum and transfected the following morning with 10 μg of *pcDNA3-Disp-Flag* per plate according to FuGene HD (Promega) instructions. Cells were incubated for an additional 48 hr prior to lysis in 1% Triton X-100 Buffer (50 mM Tris-HCl, pH 8.0, 150 mM NaCl, 1% Triton X-100, and 1X Protease Inhibitor Cocktail). Lysates were pooled and centrifuged at 14,000 x g at 4°C for 45 min. Supernatant was pre-cleared with 400 μl of 50% EZ View Red Protein A Affinity Gel (Sigma) for 1 hr at 4° C. Pre-cleared supernatant was transferred to a new tube and incubated with 400 μl of EZview Red ANTI-FLAG M2 Affinity Gel (Sigma) for 3 hr at 4° C. Beads were washed with 1% Triton X-100 Lysis Buffer with increasing amounts of NaCl (0.25M, 0.5M, 0.75M, and 0.150M) before eluting with 3x Flag Peptide (Sigma) according to instructions. Protein was TCA precipitated for 6 hr at 4°C before pelleting and resuspending in TCA Resuspension Buffer (2% w/v SDS, 0.42M Tris-HCl, pH 7.4, 4% v/v glycerol, 0.01% w/v Bromphenol Blue and 0.05M DTT). Samples were electrophoresed on a NuPage gel using NuPage MOPS running buffer (Invitrogen) then transferred to Immobilon-PSQ PVDF (Millipore) with NuPage Transfer Buffer without Methanol (Invitrogen). The PVDF membrane was stained with coomassie blue and allowed to air dry. The band of interest was excised and sent to Tufts University Core Facility (http://tucf.org/protein-f.html) for protein sequence identification.

### Drosophila phenotypic analysis

Wings from male *apterous-GAL4;UAS-disp* flies were mounted and imaged using a Zeiss ICc3 camera and processed using Adobe Photoshop. Multiple male and female progeny from at least two independent crosses were analyzed. Representative wings were imaged. For salivary gland analysis, salivary glands were dissected from *SGS-GAL4; UAS-HhGFP, UAS-V5dispHA* third instar larvae using standard methods and immunostained as described (*Carroll et al., 2012*). Disp was detected using anti-HA (1:2000) and AlexaFluor anti-mouse secondary (1:10,000, Thermofisher). Multiple salivary glands from male and female larva were examined. A representative image is shown.

## Acknowledgements

This work was supported by NIH NIGMS grants R01GM114049 and R35GM122546 and by ALSAC of St. Jude Children's Research Hospital. *Drosophila* Disp cDNA was obtained from the *Drosophila* Genomics Resource Center. Disp proteins for antibody production were purified by the SJCRH Protein Production Facility. *Drosophila* Disp antibody was generated using the Covance custom antibody service. Edman sequencing was performed at the Tufts University Proteomic Core Facility. Transgenic *Drosophila* lines were generated by Bestgene, Inc. GAL4 drivers were obtained from the Bloomington Drosophila stock center. Confocal microscopy was performed at the SJCRH Cell and Tissue Imaging Center which is supported by SJCRH and NCI P30CA021765. We thank Y Ahmed, P Beachy, A O'Reilly, H Roelink, and A Salic for reagents, M Dillard and J Kugler for technical assistance, and J Opferman, M Turnis, D Constam, D Robbins and members of the Ogden Lab for advice and discussion.

## Additional information

### Funding

| Funder | Grant reference number | Author |
|---|---|---|
| National Institute of General Medical Sciences | R01GM114049 | Stacey K Ogden |
| St. Jude Children's Research Hospital | | Stacey K Ogden |
| National Cancer Institute | P30CA021765 | Stacey K Ogden |
| National Institute of General Medical Sciences | R35GM122546 | Stacey K Ogden |

The funders had no role in study design, data collection and interpretation, or the decision to submit the work for publication.

### Author contributions

Daniel P Stewart, Data curation, Formal analysis, Validation, Investigation, Visualization, Methodology, Writing—original draft, Writing—review and editing; Suresh Marada, Data curation, Formal analysis, Validation, Investigation, Visualization, Methodology, Writing—original draft, Project administration, Writing—review and editing; William J Bodeen, Data curation, Validation, Investigation, Methodology, Writing—original draft; Ashley Truong, Data curation, Formal analysis, Validation, Investigation, Methodology; Sadie Miki Sakurada, Resources, Formal analysis, Validation, Investigation; Tanushree Pandit, Data curation, Formal analysis, Validation, Methodology; Shondra M Pruett-Miller, Resources, Data curation, Formal analysis, Methodology, Writing—review and editing; Stacey K Ogden, Conceptualization, Formal analysis, Supervision, Funding acquisition, Writing—original draft, Project administration, Writing—review and editing

### Author ORCIDs

William J Bodeen http://orcid.org/0000-0002-0557-4826
Stacey K Ogden http://orcid.org/0000-0001-8991-3065

### Decision letter and Author response

Decision letter https://doi.org/10.7554/eLife.31678.019
Author response https://doi.org/10.7554/eLife.31678.020

## Additional files

### Supplementary files

• Transparent reporting form
DOI: https://doi.org/10.7554/eLife.31678.017

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
