## [Decision Letter]

Thank you for submitting your article "Cleavage Activates Dispatched for Sonic Hedgehog Ligand Release" for consideration by *eLife*. Your article has been reviewed by three peer reviewers, and the evaluation has been overseen by a Reviewing Editor and Marianne Bronner as the Senior Editor. The reviewers have opted to remain anonymous.

The reviewers have discussed the reviews with one another and the Reviewing Editor has drafted this decision to help you prepare a revised submission.

Summary:

Your work presents a novel post-translational modification of the Dispatched protein, which is required for secretion of active Hedgehog protein. According to your findings, Dispatched is cleaved, a modification important for the function. This is an unexpected and interesting discovery that will advance our understanding of Hedgehog secretion and signaling.

Essential revisions:

After consultation among the reviewers, we ask you to address the following concerns. In addition, the full reviews are included below for your reference.

1) Try to use CRISPR/Cas9 based methods to introduce mutations at endogenous loci, instead of using RNAi (to deplete the convertases) or over-expression experiments to monitor the behavior of the DispCS. A CRISPR-mediated KO of Furin should be straightforward to do in cultured cells and will allow you to do a very feasible and definitive experiment of examining effects on endogenous Hh in somatic clones mutant for a (CRISPR?) engineered CS mutant.

2) The differences between the two Disp forms are difficult to discern on many of the blots presented in the paper, because both bands are highly smeared on gels. The quality of this data needs to be improved by either (1) using glycosidases to sharpen the bands or (2) by monitoring the shorter cleavage fragment that seems to run as a tighter band.

3) It is not clearly answered whether the cleaved form of disp is active. It has been suggested an experiment in which a truncated version corresponding to the cleaved form is expressed in a Disp null background to ensure it is capable of supporting Shh secretion. The reviewers encourage you to do this experiment.

*Reviewer #1:*

In the submitted manuscript Stewart and Marada, and colleagues describe a novel post-translational modification of Dispatched: Dispatched is cleaved and this seems to be important for its function. The observation is unexpected and interesting. It has the potential to advance our understanding of Hedgehog secretion and signaling and will inspire future research. For this reason I would suggest it is suitable for *eLife*, despite being arguably only a detailed but still incomplete analysis of a single observation.

On the whole I find that the presented data supports the conclusions made but there are some exceptions where additional data/clarification is necessary. I list these below:

In the Abstract the following statements are made; for accuracy these statements need to be corrected there and in the entire text.

"Dispatched is activated by proprotein convertase-mediated cleavage at a conserved processing site in its first extracellular loop." This is not true as in *Drosophila* – one of the models used to test the importance of the cleavage – the site is not conserved. Indeed a different mechanism is even suggested in the Discussion. A solution would be to write "vertebrate Dispatched" and delete "conserved".

"As such, convertase-mediated cleavage represents a novel regulatory process contributing to Dispatched maturation and functional competency in Hedgehog ligand producing cells." Regulatory implies that the processing is tunable/modulated – there is no evidence for this in the paper. The processing seems to be constitutive – regulatory needs to be deleted.

To be of a suitable methodological rigor the following points also need to be addressed or refuted – ideally experimentally. I hope that the data was already generated but omitted in the current version for brevity.

Figure 3

Why is Furin absent in the control sample C (lane 3)? If the result is real why is Disp not running differently?

Why does Furin knock-down not result in accumulation of Disp250?

The size of the different Dispatched forms is difficult to discern. In contrast to other experiments it looks as if there is only one form of Disp is present. Is this real?

Figure 3

The experiment should be repeated in NIH3T3 cells, or perhaps combined with PNGase F treatment to make the cleaved variants more obvious.

It is claimed that "unprocessed Disp175 enriched in Myc immunoprecipitates from lysates expressing wild type DispHA (lane 5), suggesting Furin preferentially interacted with the uncleaved Disp substrate." That the unprocessed Disp is pulldown is not obvious. A size control should be included.

Why does the expression of Furin also affect the behavior of DispCS?

Figure 3

The bands in the indicated Western blots are too blurry to claim "Conversely, murine DispHA protein expressed in LoVo cells failed to produce Disp145 (left panel). The effect was specific to Furin loss because its re-expression in LoVo cells rescued Disp145 production (right panel). Combined with the above, these results support Disp is cleaved by Furin."

DispCS should be included to show that the effect is not due to altered glycosylation.

Figure 5

How can the authors be sure that the lower band is Disp? The band seems to be unaffected by Disp dsRNA – lane 1 vs. 2. Also the size of the cleaved product following representing in lane 3 does not correspond to that seen in lane 1. Perhaps better images could be given.

Why does the Actin band run at a very different height in the DispHA Oexp lane 3?

Figure 5

To more rigorously demonstrate that DispHA overexpression (cf DispΔCS) affects Hh signaling it would be recommended to examine the expression of Hh target genes.

Figure 5

For completeness it is important to show the SGS-GAL4 UAS -hhGFP line without any Disp.

Figure 6

The presented images are uninformative and could be placed in supplementary information, or omitted.

Figure 6

Are the expression levels of the two constructs similar? A difference in the levels could explain the apparent difference in localization.

Figure 6

S2 cells are not obviously polarized; the experiment would be more informative in system described in Figure 6/E.

*Reviewer #2:*

This beautifully done study identifies a cleavage step that matures the Dispatched protein for participation in the process that releases Shh/Hh from producing cells. The analysis is based almost entirely on over-expression in cultured cells or fly tissues, relying on the assumption/inference that the observed effects are the same for endogenous Disp. I wish the authors had invested in the very do-able definitive experiment of examining effects on endogenous Hh in somatic clones mutant for a (CRISPR?) engineered CS mutant, and I strongly encourage them to do so.

*Reviewer #3:*

This manuscript by Stewart et al. tackles an important and poorly understood step in secretion of the Hedgehog ligand-regulation of the transporter-like protein Dispatched. They provide evidence for the model that Furin-mediated cleavage of Disp in the first extracellular domain (EC1) is necessary for its activity in promoting Shh secretion, perhaps by regulating its trafficking itinerary. The authors start by using Disp tagged at both ends to show cleavage and then map the cleavage site to EC1 using edman degradation. Mutation of a single residue (E208A) is used to make a non-cleavable Disp1, which is then tested in various assays to try and establish the functional importance of this cleavage reaction. Overall, this is a detailed study that uncovers a novel mechanism of Disp1 function and Shh secretion. While I am supportive of publication, there are a few important issues that should be addressed:

1) An important question is whether cleavage itself is required or whether the DispCS mutant impairs the activity of the protein in some other way. A way to show this is to prove the sufficiency of the cleavage reaction. This could be done by expression of a N-terminal truncation mutant of Disp lacking the portion of the protein from the N-terminus to the cleavage site- which could be replaced by a heterologous signal sequence. Alternatively the Furin cleavage site could be replaced by a heterologous protease cleavage site and cleavage induced by adding the cognate protease to the extracellular medium. If the authors' model is correct, this protein should be constitutively active in Shh secretion and insensitive to Furin inhibitors.

2) The siRNA mediated knockdown experiments in Figure 3 are not convincing. For instance, one of the control siRNAs (Con-C) eliminates Furin protein levels but does not impact the mobility of Disp. Also, many of the siRNAs decrease overall Disp protein levels. The blots for PACE4 and PC7 knockdowns are also not convincing. Given the widespread accessibility of genome editing technology, a clean CRISPR/Cas9-mediated knock-out of at least Furin (the pre-protein convertase with the strongest effect according to the authors) should be performed to show effects on (1) cleavage and (2) Shh secretion. The LoVo cell experiment in Figure 3 could be a substitute for this; however, an effect on Shh secretion should be demonstrated in this cell line and the lack of an effect with re-expression of catalytically dead Furin should be shown. Rather that switching to a completely different cell line (not characterized for its ability to secrete Shh in a Disp-regulated manner) for this important experiment, I would suggest making the clean CRISPR-mediated knock-out in the cell background used for most of the experiments shown in Figure 1-3.

3) In many experiments (e.g. Figure 3), we are asked to differentiate between DispHA-175 and DispHA-145 when these proteins do not appear as clearly separable bands but as partially overlapping smears. One concern is that these differences are simply due to glycosylation differences and do not reflect cleaved and uncleaved species. It would be much cleaner to monitor the production of the ~37kDA N-terminal fragment in these experiments.

---

## [Author Response]

Essential revisions:After consultation among the reviewers, we ask you to address the following concerns. In addition, the full reviews are included below for your reference.1) Try to use CRISPR/Cas9 based methods to introduce mutations at endogenous loci, instead of using RNAi (to deplete the convertases) or over-expression experiments to monitor the behavior of the DispCS. A CRISPR-mediated KO of Furin should be straightforward to do in cultured cells and will allow you to do a very feasible and definitive experiment of examining effects on endogenous Hh in somatic clones mutant for a (CRISPR?) engineered CS mutant.

We used CRISPR/Cas9 to knock out *Furin, Pace4, Pcsk5* and *Pcsk7* in mouse embryonic fibroblasts (MEFs) derived from C57BL/6 mice. We show that MEF cells genetically null for *Furin* do not cleave Dispatched, but *Pace4, Pcsk5*, and *Pcsk7* knockout cells do. These results are shown in new Figure 3, and replace the siRNA experiment provided in original Figure 3. Cleavage experiments were performed using two clonal lines for each gene knockout. Results from the second clonal line are shown in Figure 3—figure supplement 1. Antibodies against PACE4 and PCSK5 did not detect the endogenous proteins in control MEF cells, so we provide deep sequencing data validating the mutations in Figure 3—figure supplement 1.

We tested the ability of *Furin* knockout cells to release Shh ligand into conditioned media. Although these cells were not healthy, making them difficult to transfect, we were able to perform transient transfections to assess Shh release. We show that whereas control MEFs do release Shh, *Furin* knockout cells fail to effectively release ligand into culture media. These results are shown in Figure 4.

2) The differences between the two Disp forms are difficult to discern on many of the blots presented in the paper, because both bands are highly smeared on gels. The quality of this data needs to be improved by either (1) using glycosidases to sharpen the bands or (2) by monitoring the shorter cleavage fragment that seems to run as a tighter band.

We repeated many of the original experiments to monitor formation of the ~30 kDa cleavage fragment. We now show the lower band in new Figure panels 1D, 2D-E, and 3C-E. Monitoring the shorter fragment does indeed improve clarity of the results. We thank the reviewers for this helpful suggestion.

3) It is not clearly answered whether the cleaved form of disp is active. It has been suggested an experiment in which a truncated version corresponding to the cleaved form is expressed in a Disp null background to ensure it is capable of supporting Shh secretion. The reviewers encourage you to do this experiment.

We tried this experiment several times, but have not obtained informative results due to Dispatched amino-terminal truncation proteins being unstable and/or retained in the ER.

A) An amino-terminal truncation mutant corresponding to the cleaved form of Dispatched (Disp∆N) was generated. This mutant was unstable when expressed in NIH3T3 cells, perhaps due to loss of the signal sequence, which is predicted to be in the first transmembrane domain.

B) We next generated a chimeric mutant in which the first transmembrane domain of Wntless, the protein responsible for transport of Wnt ligands to the cell surface, was introduced into Disp∆N. We reasoned that by replacing the Dispatched first transmembrane domain with a Wls domain encoding a signal sequence, it might get to the cell surface. Although this mutant was more stable than Disp∆N, it was retained in the ER.

C) We introduced various plasma membrane targeting sequences onto Disp∆N. These included the N-terminal plasma membrane targeting sequence of the kinase Lck as in Klapisz et al., Eur J Biochem, 1999 Nov; 265(3):957, and the plasma membrane targeting signal of K-ras4B with a polybasic sequence as used in DiPilato et al., PNAS, 2004;101(47):16513. Both of these Disp∆N proteins were retained in the ER, and were significantly less stable than full-length Dispatched.

As would be predicted based on their failure to traffic to the plasma membrane, none of these mutants showed functional activity in co-culture reporter assays. We have not included these negative data in the revised manuscript, but will include the results if the Editor and reviewers think it necessary. We have added a sentence to the sixteenth paragraph of the Results, stating that although our results suggest that Disp145 is the active form of the protein, we were unable to test its activity directly due to our inability to get an amino-terminally truncated Disp protein to the cell surface.

Additional Revisions

Some additional revisions suggested by the reviewers have been incorporated in the current submission, as outlined below:

- We included text revisions throughout the manuscript as suggested by reviewers 1 and 2.

- We removed the uninformative wing imaginal disc result that was originally shown as Figure 6, as suggested by reviewers 1 and 2.

- We included a control salivary gland of an *SGS-GAL4;UAS-hhGFP* animal as Figure 5, as requested by reviewer 1. The control salivary gland that lacks Dispatched (Figure 5) accumulates HhGFP at a level similar to the DispCS-expressing salivary gland (Figure 5), supporting that DispCS is nonfunctional in vivo.

- We removed the BFA experiment originally shown as Figure 1, as suggested by reviewer 3. The new cell surface biotinylation experiment and quantification shown in revised Figure 1 more clearly demonstrate that processing occurs on the cell surface, eliminating the need for including the BFA results.